# EFFICIENT DISTRIBUTION MATCHING OF REPRESENTATIONS VIA NOISE-INJECTED DEEP INFOMAX

**Ivan Butakov**[*,1,2], **Alexander Semenenko**[1], **Alexander Tolmachev**[1,2], **Andrey Gladkov**[1],
**Marina Munkhoeva**[3], **Alexey Frolov**[1]
[1]Skolkovo Institute of Science and Technology; [2]Moscow Institute of Physics and Technology;
[3]Artificial Intelligence Research Institute;
{butakov.id, semenenko.av, tolmachev.ad, gladkov.ao}@phystech.su,
munkhoeva@airi.net, al.frolov@skoltech.ru

## ABSTRACT

Deep InfoMax (DIM) is a well-established method for self-supervised representation learning (SSRL) based on maximization of the mutual information between the input and the output of a deep neural network encoder. Despite the DIM and contrastive SSRL in general being well-explored, the task of learning representations conforming to a specific distribution (i.e., distribution matching, DM) is still under-addressed. Motivated by the importance of DM to several downstream tasks (including generative modeling, disentanglement, outliers detection and other), we enhance DIM to enable automatic matching of learned representations to a selected prior distribution. To achieve this, we propose injecting an independent noise into the normalized outputs of the encoder, while keeping the same InfoMax training objective. We show that such modification allows for learning uniformly and normally distributed representations, as well as representations of other absolutely continuous distributions. Our approach is tested on various downstream tasks. The results indicate a moderate trade-off between the performance on the downstream tasks and quality of DM.

## 1 INTRODUCTION

Learning viable low-dimensional representations of complex data plays an important role in many modern applications of artificial intelligence. This task arises in various domains, including image (Haralick et al., 1973; Chen et al., 2020b; Rombach et al., 2022), audio (van den Oord et al., 2019), and natural language processing (Mikolov et al., 2013; Radford et al., 2018; Devlin et al., 2019). High-quality embeddings are particularly useful for multi-modal methods (Vinyals et al., 2015; Radford et al., 2021; Ho & Salimans, 2021), statistical and topological analysis (Moor et al., 2020; Duong & Nguyen, 2022; Butakov et al., 2024b), data visualization (van der Maaten & Hinton, 2008; McInnes et al., 2018), and testing fundamental hypotheses (Brown et al., 2023; Gurnee & Tegmark, 2024; Huh et al., 2024).

Existing approaches to representation learning can be divided into three categories (Ericsson et al., 2022): *supervised* (requires labeled data), *self-supervised* and *unsupervised* (no labeling is required). In practice, access to labeled data is limited, which hinders the use of supervised approaches. Therefore, unsupervised and self-supervised methods are of great importance. Contrastive learning is a well-established paradigm of self-supervised representation learning (SSRL), which encourages an encoder to learn similar representations for various augmentations of the same data point, and dissimilar – for different data points. Deep InfoMax (DIM) (Hjelm et al., 2019) leverages information-theoretic quantities to construct a decent contrastive objective, involving a direct maximization of the useful information contained in the embeddings. DIM is universal, flexible, and deeply connected to the rigorous information theory, which allows for a good performance on a variety of downstream tasks to be attained (Hjelm et al., 2019; Bachman et al., 2019; Veličković et al., 2019; Tschannen et al., 2020; Yu, 2024).

---

*Correspondence to butakov.id@phystech.edu

Acquiring embeddings admitting a specific distribution (i.e., distribution matching, DM) is an auxiliary, yet important task in representation learning. Latent distributions with straightforward sampling procedures or tractable densities are crucial for downstream generative modeling (Kingma & Welling, 2014; Makhzani et al., 2016; Larsen et al., 2016; Papamakarios et al., 2021). Additionally, specific distributions (e.g., Gaussian) exhibit properties, which are useful for statistical analysis (Tipping & Bishop, 1999; Duong & Nguyen, 2022), disentanglement (Higgins et al., 2017; Balabin et al., 2024), and outliers detection (Bazarova et al., 2024).

A classical approach to latent DM is to optimize a cheap and imprecise distribution dissimilarity measure during the training or architecture search (Ng, 2011; Makhzani & Frey, 2014; Kingma & Welling, 2014; Heusel et al., 2017; Higgins et al., 2017). Methods of this family have to rely on several strong assumptions, such as embeddings already admitting a Gaussian distribution, which eventually leads to suboptimal results. Another common approach employs adversarial networks to push the learned representations towards a desired distribution (Makhzani et al., 2016; Hjelm et al., 2019). One can also leverage generative models (e.g., normalizing flows or diffusion models) to explicitly perform DM "post hoc" (Böhm & Seljak, 2022; Rombach et al., 2022). These two approaches yield decent results, but require supplementary networks to match the distributions. Finally, injective likelihood-based models can also be used to acquire low-dimensional representations admitting a desired distribution (Brehmer & Cranmer, 2020; Sorrenson et al., 2024). However, likelihood maximization across dimensions is notoriously problematic due to non-square Jacobi matrices.

In contrast to the mentioned approaches, we propose a simple, cost-effective (requiring no additional neural networks) and non-intrusive modification to DIM, which allows for an automatic and exact DM of the representations. In the following text, we show that using specific activation functions and noise injections at the outlet of an encoder, combined with the DIM objective, allows for normally and uniformly distributed representations to be learned. Our contributions are the following:

1. We prove that applying normalization and adding a small noise at the outlet of an encoder and maximizing the DIM objective minimizes the Kullback-Leibler divergence between a Gaussian (or uniform) distribution and the distribution of embeddings.

2. We conduct experiments on several downstream tasks to explore the trade-off between the downstream performance and the accuracy of DM via our method.

3. We conduct additional experiments to assess the quality of DM via our method in the tasks of generative modelling.

The paper is organized as follows. In Section 2, the necessary background from information theory and original works on Deep InfoMax is provided. In Section 3, we describe the general method for DM via modified DIM. In Section 4, a connection between the proposed approach and other SSRL methods is established. Section 5 is dedicated to the experimental evaluation of our method. Finally, we conclude the paper by discussing our results in Section 6. Complete proofs, additional analysis of infomax-based DM, and technical details are provided in Appendices A to C correspondingly.

## 2 BACKGROUND

In this section, the background necessary to understand our work is provided. We start with the basic definitions from the information theory. Then, the maximum entropy theorems are given, which are crucial for understanding our approach. Finally, we outline the general variant of Deep InfoMax representation learning method, which we aim to enhance with an automatic distribution matching.

### 2.1 PRELIMINARIES

Let $(\Omega, \mathcal{F}, \mathbb{P})$ be a probability space with sample space $\Omega$, $\sigma$-algebra $\mathcal{F}$, and probability measure $\mathbb{P}$ defined on $\mathcal{F}$. Consider an absolutely continuous random vector $X \colon \Omega \to \mathbb{R}^d$ with the probability density function (PDF) denoted as $p(x)$. The differential entropy of $X$ is defined as follows:

$$h(X) = -\mathbb{E} \log p(X) = -\int_{\text{supp} X} p(x) \log p(x)\, dx,$$

where $\operatorname{supp} X \subseteq \mathbb{R}^d$ represents the *support* of $X$, and $\log(\,\cdot\,)$ denotes the natural logarithm. Similarly, we define the joint differential entropy as $h(X, Y) = -\mathbb{E} \log p(X, Y)$ and conditional differential entropy as $h(X \mid Y) = -\mathbb{E} \log p(X|Y) = -\mathbb{E}_Y\left(\mathbb{E}_{X|Y} \log p(X \mid Y)\right)$. Finally, the mutual information (MI) is given by $I(X; Y) = h(X) - h(X \mid Y)$, and the following equivalences hold

$$I(X; Y) = h(X) - h(X \mid Y) = h(Y) - h(Y \mid X),$$

$$I(X; Y) = h(X) + h(Y) - h(X, Y),$$

$$I(X; Y) = \mathrm{D_{KL}}\left(\mathbb{P}_{X,Y} \,\|\, \mathbb{P}_X \otimes \mathbb{P}_Y\right).$$

Mutual information can also be defined as an expectation of the *pointwise mutual information*:

$$\mathrm{PMI}_{X,Y}(x, y) = \log\left[\frac{p(x \mid y)}{p(x)}\right], \quad I(X; Y) = \mathbb{E} \, \mathrm{PMI}_{X,Y}(X, Y). \tag{1}$$

The above definitions can be generalized via Radon-Nikodym derivatives and induced densities in case of distributions supports being manifolds, see (Spivak, 1965). We use flexible notation, where entropy and divergence may refer to random vectors or their corresponding distributions.

Our work leverages the maximum entropy properties of Gaussian and uniform distributions:

**Theorem 2.1** (Theorem 8.6.5 in Cover & Thomas (2006))**.** *Let $X$ be a $d$-dimensional absolutely continuous random vector with probability density function $p$, mean $m$ and covariance matrix $\Sigma$. Then*

$$h(X) = h\left(\mathcal{N}(m, \Sigma)\right) - \mathrm{D_{KL}}\left(p \,\|\, \mathcal{N}(m, \Sigma)\right), \qquad h\left(\mathcal{N}(m, \Sigma)\right) = \frac{1}{2} \log\left((2\pi e)^d \det \Sigma\right),$$

*where $\mathcal{N}(m, \Sigma)$ is a Gaussian distribution of mean $m$ and covariance matrix $\Sigma$.*

**Theorem 2.2.** *Let $X$ be an absolutely continuous random vector with probability density function $p$ and $\operatorname{supp} X \subseteq S$, where $S$ has finite and non-zero Lebesgue measure $\mu(S)$. Then*

$$h(X) = h\left(\mathrm{U}(S)\right) - \mathrm{D_{KL}}\left(p \,\|\, \mathrm{U}(S)\right), \qquad h\left(\mathrm{U}(S)\right) = \log \mu(S),$$

*where $\mathrm{U}(S)$ is a uniform distribution on $S$.*

*Remark* 2.3. Note that $h(X) \neq h(X') - \mathrm{D_{KL}}\left(p_X \,\|\, p_{X'}\right)$ in general.

Finally, we also utilize the following lower bound on the conditional entropy of a sum of two conditionally independent random vectors:

**Lemma 2.4.** *Let $X$ and $Z$ be random vectors of the same dimensionality, independent under the conditioning vector $Y$. Then*

$$h(X + Z \mid Y) = h(Z \mid Y) + I(X; X + Z \mid Y) \geq h(Z \mid Y),$$

*with equality if and only if there exists a measurable function $g$ such that $X = g(Y)$.*[1]

## 2.2 DEEP INFOMAX

Mutual information (MI) is widely considered as a fundamental measure of statistical dependence between random variables due to its key properties, such as invariance under diffeomorphisms, sub-additivity, non-negativity, and symmetry (Cover & Thomas, 2006). These attributes make MI particularly useful in information-theoretic approaches to machine learning. The concept of maximizing MI between input and output, known as the infomax principle (Linsker, 1988; Bell & Sejnowski, 1995), serves as the foundation for Deep InfoMax (DIM), a family of self-supervised representation learning methods proposed by Hjelm et al. (2019).

A naïve DIM approach suggests learning the most informative embeddings via a direct maximization of the mutual information between the original data and compressed representations:

$$I(X; f(X)) \to \max,$$

---

[1] Hereinafter, when comparing random variables, we mean equality or inequality "almost sure".

where $X$ is the random vector to be compressed, and $f$ is the encoding mapping (being learned). However, despite its simplicity and intuitiveness, this setting is rendered useless by the fact that $I(X; f(X)) = \infty$ for a wide range of $X$ and $f$ (Bell & Sejnowski, 1995; Hjelm et al., 2019).

To avoid this limitation, it is usually suggested to augment (crop randomly, add noise, etc.) original data to inject stochasticity and make information-theoretic objectives non-degenerate. Consider the following Markov chain:

$$f(X) \longrightarrow X \longrightarrow X',$$

where $X'$ is augmented data. Now $I(X'; f(X))$ can be made non-infinite. Moreover, according to the data processing inequality (Cover & Thomas, 2006), $I(X'; f(X)) \leq I(X; f(X))$, which connects this new infomax objective with the naïve one.

The only remaining problem is the high dimensionality of $X'$, which makes MI estimation difficult. To resolve this, one can apply an additional (in general, random) dimensionality-reducing transformation to replace $X'$ with $Y'$:

$$f(X) \longrightarrow X \longrightarrow X' \longrightarrow Y',$$

with $I(Y'; f(X)) \leq I(X'; f(X))$ being the new infomax objective. For the sake of computational simplicity, $Y'$ can be defined as $f(X')$; thus, the same encoder network is used to compress both the original and augmented data. Moreover, one can view $I(f(X'); f(X))$ as a contrastive objective: this value is high when $f$ yields similar representations for corresponding augmented and non-augmented samples, and dissimilar for other pairs of samples.

Now, let us consider the task of mutual information maximization. As MI is notoriously hard to estimate (McAllester & Stratos, 2020), lower bounds are widely used to reparametrize the infomax objective (Belghazi et al., 2021; van den Oord et al., 2019; Hjelm et al., 2019). In this work, we only consider the two most popular approaches based on the variational representations of the Kullback-Leibler divergence:

*Donsker-Varadhan*
(Donsker & Varadhan 1983
Belghazi et al. 2021)
$$I(X; Y) = \sup_{T \colon \Omega \to \mathbb{R}} \left[ \mathbb{E}_{\mathbb{P}_{X,Y}} T - \log \mathbb{E}_{\mathbb{P}_X \otimes \mathbb{P}_Y} \exp(T) \right] \quad (2)$$

*Nguyen-Wainwright-Jordan*
(Nguyen et al. 2010,
Belghazi et al. 2021)
$$I(X; Y) = \sup_{T \colon \Omega \to \mathbb{R}} \left[ \mathbb{E}_{\mathbb{P}_{X,Y}} T - \mathbb{E}_{\mathbb{P}_X \otimes \mathbb{P}_Y} \exp(T - 1) \right] \quad (3)$$

where $\Omega$ is the sampling space, and $T$ is a measurable critic function.

*Remark* 2.5. Assuming $\mathrm{PMI}_{X,Y}$ exists, the supremum in (2) and (3) is attained at and only at

$$T^* = T^*(x, y) = \mathrm{PMI}_{X,Y}(x, y) + \alpha \quad \text{(a.s. w.r.t. } \mathbb{P}_{X,Y}), \quad (4)$$

where $\alpha = 1$ for (3) and can be any real number for (2) (follows from Theorem 1 in Belghazi et al. (2021) and Theorem 2.1 in Keziou (2003)).

In practice, $f$ and $T$ are approximated via corresponding neural networks, with the parameters being learned through the maximization of the Monte-Carlo estimate of (2) or (3). Although (2) usually yields better estimates, the SGD gradients of this expression are biased in a mini-batch setting (Belghazi et al., 2021).

## 3 DEEP INFOMAX WITH AUTOMATIC DISTRIBUTION MATCHING

In the present section, we modify DIM to enable automatic distribution matching (DM) of representations. Specifically, we propose adding independent noise $Z$ to normalized representations of $X$, thus replacing $f(X)$ by $f(X) + Z$ in the infomax objective from Section 2.2. This corresponds to the following Markov chain:

$$f(X) + Z \longrightarrow f(X) \longrightarrow X \longrightarrow X' \longrightarrow f(X').$$

By the data processing inequality, we know that $I(f(X'); f(X) + Z) \leq I(f(X'); f(X))$, which connects our method to the family of conventional DIM approaches discussed previously.

As we will demonstrate, the noise injection at the outlet of an encoder, in conjunction with Theorems 2.1 and 2.2 on maximum entropy, allows us to achieve normally or uniformly distributed

embeddings. To prove this, we use the following Lemma 3.1, which decomposes the proposed in-fomax objective into three components: the entropy of representations with an additive noise $Z$, the entropy of the noise itself, and the mutual information between $f(X)$ and $f(X) + Z$ conditioned on $f(X')$ – representations of augmented data.

**Lemma 3.1.** *Consider the following Markov chain of absolutely continuous random vectors:*

$$f(X) + Z \longrightarrow X \longrightarrow X' \longrightarrow f(X'),$$

*with $Z$ being independent of $(X, X')$. Then*

$$I(f(X'); f(X) + Z) = h(f(X) + Z) - h(Z) - I(f(X) + Z; f(X) \mid f(X')). \qquad (5)$$

Lemma 3.1 highlights the importance of both additive noise and input data augmentation. Specifically, if no augmentation is applied, i.e., $X = X'$, then $I(f(X) + Z; f(X) \mid f(X')) = 0$. In this case, maximizing the infomax objective is reduced to maximizing the entropy $h(f(X) + Z)$. Under the corresponding restrictions on $f(X)$, this is equivalent to distribution matching, see Theorems 2.1 and 2.2. In contrast, if no noise is added, $I(f(X'); f(X))$ is not bounded in general, which may prevent $h(f(X))$ from saturation in practical scenarios.

Additionally, DM alone does not guarantee that the learned representations will be meaningful or useful for downstream tasks. That is why we also show that using $X \neq X'$ allows us to recover meaningful embeddings, as $I(f(X) + Z; f(X) \mid f(X'))$ is set to zero by learning representations that are *weakly invariant* to selected data augmentations:

**Definition 3.2.** We call an encoding mapping $f$ *weakly invariant* to data augmentation $X \to X'$ if there exists a function $g$ such that $f(X) = g(f(X)) = g(f(X'))$ almost surely.

**Lemma 3.3.** *Under the conditions of Lemma 3.1, let $\mathbb{P}(X = X' \mid X) \geq \alpha > 0$. Then, $I(f(X) + Z; f(X) \mid f(X')) = 0$ precisely when $f$ is weakly invariant to $X \to X'$.*

Overall, the signal-to-noise ratio serves as a tradeoff between the distribution matching objective and robustness to the data augmentations (the first and the third term in (5) correspondingly): higher magnitudes of $Z$ impose tighter bounds on $I(f(X) + Z; f(X) \mid f(X'))$, thus prioritizing maximization of $h(f(X) + Z)$. In what follows, we formalize the provided reasoning by addressing Gaussian and uniform distribution matching separately. We also briefly discuss how a general DM problem can be reduced to the normal or uniform one.

## 3.1 GAUSSIAN DISTRIBUTION MATCHING

Let us assume $Z$ having finite second-order moments. According to Theorem 2.1, if we restrict $f(X)$ by fixing its covariance matrix, $h(f(X) + Z)$ attains its maximal value precisely when $f(X) + Z$ is distributed normally. Now, as $Z$ is independent of $X$, $f(X) + Z$ is normally distributed if and only if the distribution of $f(X)$ is also Gaussian.

Thus, one can achieve the normality of $f(X)$ via restricting second-order moments of $f(X)$ and maximizing the entropy $h(f(X) + Z)$. In combination with Lemma 3.1, this forms a basis of the proposed method for Gaussian distribution matching. Moreover, we show that the imposed restrictions can be partially lifted via only requiring $\text{Var}(f(X)_i) = 1$ for every $i \in \{1, \ldots, d\}$. This approach is preferable in practice, as the widely-used batch normalization (Ioffe & Szegedy, 2015) can be employed to restrict the variances. We formalize our findings in the following theorem.

**Theorem 3.4** (Gaussian distribution matching). *Let the conditions of Lemma 3.3 be satisfied. Assume $Z \sim \mathcal{N}(0, \sigma^2 \mathrm{I})$, $\mathbb{E} f(X) = 0$ and $\text{Var}(f(X)_i) = 1$ for all $i \in \{1, \ldots, d\}$. Then, the mutual information $I(f(X'); f(X) + Z)$ can be upper bounded as follows*

$$I(f(X'); f(X) + Z) \leq \frac{d}{2} \log\left(1 + \frac{1}{\sigma^2}\right), \qquad (6)$$

*with the equality holding exactly when $f$ is weakly invariant and $f(X) \sim \mathcal{N}(0, \mathrm{I})$. Moreover,*

$$\mathrm{D}_{\mathrm{KL}}\left(f(X) \,\|\, \mathcal{N}(0, \mathrm{I})\right) \leq I(Z; f(X) + Z) - I(f(X'); f(X) + Z) - d \log \sigma.$$

While the first inequality in Theorem 3.4 suggests that exact distribution matching is achieved only asymptotically, the second expression shows that $\mathrm{D}_{\mathrm{KL}}\left(f(X) \,\|\, \mathcal{N}(0, \mathrm{I})\right)$ can be bounded using the value of the proposed infomax objective. Note that $I(Z; f(X) + Z)$ is typically upper bounded, except the rare cases where the support of $f(X)$ is degenerate.

## 3.2 UNIFORM DISTRIBUTION MATCHING

Now consider $Z$ distributed uniformly on $[-\varepsilon; \varepsilon]^d$. We assume that $f(X)$ has bounded support within $[0, 1]^d$. This restriction serves as an alternative to fixing the second-order moments, as in the Gaussian case, and is easy to implement in practice (e.g., via a sigmoid function). By Theorem 2.2, the entropy $h(f(X) + Z)$ is maximized when $f(X) + Z$ follows a uniform distribution. This can be achieved when $f(X)$ conforms to a discrete uniform distribution over an equidistant grid within $[0, 1]^d$. As $\varepsilon$ approaches zero, the distribution of $f(X)$ gradually converges to a continuous uniform distribution over $[0, 1]^d$. Thus, injecting uniform noise asymptotically drives $f(X)$ toward a uniform representation over $[0, 1]^d$. This reasoning is formalized in the following theorem.

**Theorem 3.5** (Uniform distribution matching). *Under the conditions of Lemma 3.1, let $Z \sim U([-\varepsilon; \varepsilon]^d)$ and $\operatorname{supp} f(X) \subseteq [0; 1]^d$. Then, the mutual information $I(f(X'); f(X) + Z)$ can be upper bounded as follows*

$$I(f(X'); f(X) + Z) \leq d \log \left( 1 + \frac{1}{2\varepsilon} \right), \tag{7}$$

*with the equality if and only if $1/\varepsilon \in \mathbb{N}$, $f$ is weakly invariant, and $f(X) \sim U(A)$, where the set $A = \{0, 2\varepsilon, 4\varepsilon, \ldots, 1\}$ contains $(1/(2\varepsilon) + 1)$ elements.*

*Moreover,*

$$D_{KL} \left( f(X) \parallel U([0; 1]^d) \right) \leq I(Z; f(X) + Z) - I(f(X'); f(X) + Z) - d \log(2\varepsilon).$$

In sharp contrast to Theorem 3.4, the equality in (7) is attained at $f(X)$ conforming to a discrete distribution. This makes the proposed objective less attractive in comparison to the Gaussian distribution matching. However, one still can be assured that the standard continuous uniform distribution allows us to approach the equality in (7) with $\varepsilon$ approaching zero:

*Remark* 3.6 (Butakov et al. 2024a). If $f(X) \sim U([0; 1]^d)$, $f$ is weakly invariant, and $\varepsilon < 1/2$, then

$$I(f(X'); f(X) + Z) = d(\varepsilon - \log(2\varepsilon)) = d \log(1 + 1/(2\varepsilon)) - \underbrace{(\log(1 + 2\varepsilon) - \varepsilon)}_{o(\varepsilon)}.$$

## 3.3 GENERAL CASE

Our approach can be extended to a wide range of desired distributions of embeddings using the probability integral transform (David & Johnson, 1948; Chen & Gopinath, 2000). Normalizing flows (learnable diffeomorphisms) can also be leveraged to transform the distribution "post-hoc" due to their universality property (Huang et al., 2018; Jaini et al., 2019; Kobyzev et al., 2020).

The data processing inequality can be used to estimate the quality of DM after the transformation:

**Statement 3.7** (Corollary 2.18 in (Polyanskiy & Wu, 2024)). *Let $g$ be a measurable (w.r.t $\mathbb{P}$ and $\mathbb{Q}$) function. Then $D_{KL} \left( \mathbb{P} \circ g^{-1} \parallel \mathbb{Q} \circ g^{-1} \right) \leq D_{KL} \left( \mathbb{P} \parallel \mathbb{Q} \right)$, where $\mathbb{P} \circ g^{-1}$ and $\mathbb{P} \circ g^{-1}$ denote the push-forward measures of $\mathbb{P}$ and $\mathbb{Q}$ after applying $g$.*

## 4 CONNECTION TO OTHER METHODS FOR SSRL

In this section, we explore the relation of our approach to other methods for unsupervised and self-supervised representation learning. We argue that the proposed technique for distribution matching can be applied to any other method, given the latter can be formulated in terms of mutual information maximization.

**Autoencoders** In reconstruction-based generative models (Makhzani et al., 2016), the reconstruction error is closely tied to the mutual information (Theorem 8.6.6 in (Cover & Thomas, 2006)):

$$I(X; Y) = h(X) - h(X \mid Y) \geq h(X) - \frac{d}{2} \log \left( \frac{2\pi e}{d} \right) - \frac{d}{2} \log \mathbb{E}[\|X - \hat{X}(Y)\|^2].$$

Here, $X$ denotes the input, $Y$ the latent representation produced by the encoder, and $\hat{X}(Y)$ the reconstructed input. The last term involving the expected squared error[2] is essentially the autoencoder's loss function. By minimizing this reconstruction loss, the mutual information between the

---

[2]Here and throughout, $\| \cdot \|$ denotes the Euclidean norm.

input data and its representation is maximized. However, recall that $I(X;Y)$ can diverge to infinity, as discussed earlier, so it is crucial to introduce augmentations.

**InfoNCE** Many conventional methods for self-supervised representation learning leverage InfoNCE loss (van den Oord et al., 2019) in conjunction with a similarity measure $T(\cdot, \cdot)$ to formulate the following contrastive objective (Tschannen et al., 2020; He et al., 2020; Chen et al., 2020a):

$$\mathcal{L}_{\text{InfoNCE}} = -\left[\mathbb{E}_{\mathbb{P}^+} T(Q, K) - \log \mathbb{E}_{\mathbb{P}^-} \exp(T(Q, K))\right], \qquad \hat{\mathcal{L}}_{\text{InfoNCE}} = -\log \frac{e^{T(q,k^+)}}{\frac{1}{K}\sum_{i=1}^K e^{T(q,k_i^-)}},$$

where $\mathbb{P}^+$ and $\mathbb{P}^-$ denote the distributions of positive and negative pairs of keys $K$ and queries $Q$ correspondingly. By selecting $\mathbb{P}^+ = \mathbb{P}_{f(X'),f(X)+Z}$ and $\mathbb{P}^- = \mathbb{P}_{f(X')} \otimes \mathbb{P}_{f(X)+Z}$, we recover the Donsker-Varadhan bound (2) for our infomax objective $I(f(X'); f(X) + Z)$. However, note that in (2) and (3) the supremum is taken over all measurable functions. In contrast, non-infomax methods typically employ a *separable critic*: $T(q, k) = \langle \phi(q), \psi(k) \rangle$, where $\phi$ and $\psi$ are *projection heads* (Tschannen et al., 2020; Chen et al., 2020a). In special cases $\phi, \psi = \text{Id}$, so similarity of representations is measured via a plain dot product.

Despite this significant difference, our distribution matching paradigm allows us to drop the supremum in (2) and (3) and establish a direct connection between DIM and traditional non-infomax contrastive SSRL methods:

**Theorem 4.1** (Dual form of Gaussian distribution matching). *Under the conditions of Theorem 3.4,*

$$I(f(X'); f(X) + Z) \geq \mathbb{E}_{\mathbb{P}^+}\left[T^*_{\mathcal{N}(0,\sigma^2 \text{I})}\right] - \log \mathbb{E}_{\mathbb{P}^-}\left[\exp\left(T^*_{\mathcal{N}(0,\sigma^2 \text{I})}\right)\right],$$

$$T^*_{\mathcal{N}(0,\sigma^2 \text{I})}(x, y) = \frac{\|y\|^2}{2(1+\sigma^2)} - \frac{\|y-x\|^2}{2\sigma^2} = \frac{1}{\sigma^2}\left(\langle x, y \rangle - \frac{\|x\|^2 + \|y\|^2/(1+\sigma^2)}{2}\right),$$

*with the equality holding precisely when $f$ is weakly invariant and $f(X) \sim \mathcal{N}(0, \text{I})$.*

Note that $\langle x, y \rangle$ is widely used as a similarity measure, $\sigma^2$ can be interpreted as temperature, and the remaining part of the expression serves as a regularization term.

**Covariance-based methods** Barlow Twins (Zbontar et al., 2021) and VICReg (Bardes et al., 2021) objective functions can be traced to information-theoretic terms either through Information Bottleneck (Tishby et al., 1999) or multi-view InfoMax (Federici et al., 2020) principles:

$$h(f(X') \mid X) - \lambda h(f(X')) \to \min \quad \text{(BarlowTwins)}$$

$$I(f(X'); X') \geq h(f(X')) + \mathbb{E}[\log q(f(X') \mid X'')] \to \max \quad \text{(VICReg)}$$

where $\lambda > 0$, $X \to X''$ is a separate augmentation path, independent of $X \to X'$, and $q$ is a probability density function of some distribution. In both cases, the representation entropy $h(f(X'))$ comes into play. Recall that normal distribution is the maximum entropy distribution given first two moments (Theorem 2.1). As both methods employ covariance restriction terms in the respective objective functions, one can recover these methods by assuming the normality of $f(X')$.

## 5 EXPERIMENTS

In this section, we evaluate our distribution matching approach on several datasets and downstream tasks. To assess the quality of the embeddings, we solve downstream classification tasks and calculate clustering scores. To explore the relation between the magnitude of injected noise and the quality of DM, a set of statistical normality tests is employed. For the experiments requiring numerous evaluations or visualization, we use MNIST handwritten digits dataset LeCun et al. (2010). For other experiments, we use CIFAR-10, CIFAR-100 datasets (Krizhevsky, 2009) and ImageNet (Russakovsky et al., 2015).

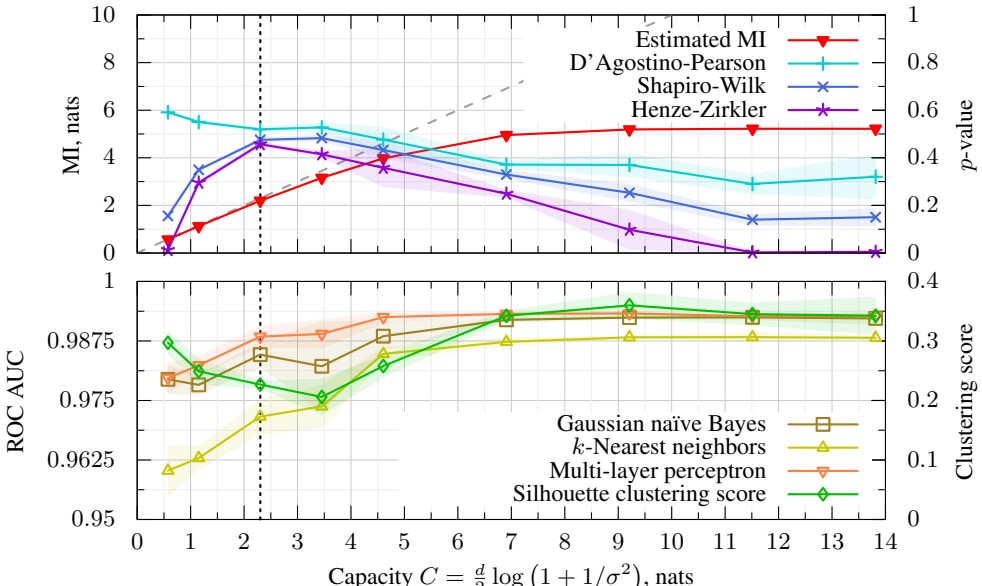

Figure 1: Results for MNIST dataset in the Gaussian DM setup for $d = 2$ with varying capacity $C = \frac{d}{2} \log \left(1 + 1/\sigma^2\right)$, measured in nats (units of information based on natural logarithms) . The dotted line denotes the minimal capacity required to preserve the information about the class labels in $f(X) + Z$. The dashed line represents the upper bound on the mutual information (6). We run 5 experiments for each point and report mean values and 99% asymptotic confidence intervals. InfoNCE loss is used to approximate (2).

**Multivariate normality and uniformity tests**  The key part of the experimental pipeline is to estimate how much the distribution of embeddings acquired via the proposed method is similar to the multivariate normal or uniform distribution. To do this, we leverage D'Agostino-Pearson (D'Agostino, 1971; D'Agostino & Pearson, 1973), Shapiro-Wilk (González-Estrada et al., 2022) univariate tests, and Henze-Zirkler (Henze & Zirkler, 1990; Trujillo-Ortiz et al., 2007) multivariate test. To extend the univariate tests to higher dimensions, we utilize the fundamental property of the Gaussian distribution, which is the normality of any linear projection.

In practice, we sample random projectors from a $d$-dimensional sphere to perform a univariate normality test. We also employ bootsrapping with small subsampling size to get low-variance averaged $p$-values and to smooth the transition from low to high $p$-values. Finally, we reduce the uniform distribution case to the Gaussian via the probability integral transform, see Section 3.3. We report the results of the tests in Figure 1. We also visualize the two-dimensional embeddings of the MNIST dataset in Figure 2. Essentially similar experiments are also conducted on CIFAR-10 dataset in Appendix E.

**Classification and clustering**  To explore the trade-off between the magnitude of the injected noise and the quality of representations, we evaluate various clustering metrics and perform downstream classification using conventional ML methods, such as Gaussian naïve Bayes, $k$-nearest neighbors and shallow multilayer perceptron. In order to numerically measure the quality of clustering, we compute Silhouette score (Rousseeuw, 1987). The results are reported in Figure 1.

We also verify noise injection does not affect typical methods for SSRL such as SimCLR (Chen et al., 2020a) and VICReg (Bardes et al., 2021). To this end, we train both methods on CIFAR-10 and CIFAR-100 datasets with varying degree of noise. The linear probing performance only slightly drops when noise magnitude is increased across $\sigma$ values $(0.0, 0.1, 0.3, 0.5)$ as seen in Table 1. Finally, to validate this behavior on a large-scale dataset, we pre-train VICReg with noise injection on ImageNet-100 and ImageNet-1k during 100 epochs. Linear probing results on respective datasets are reported in Table 2 below. See Appendix F for implementation and training details.

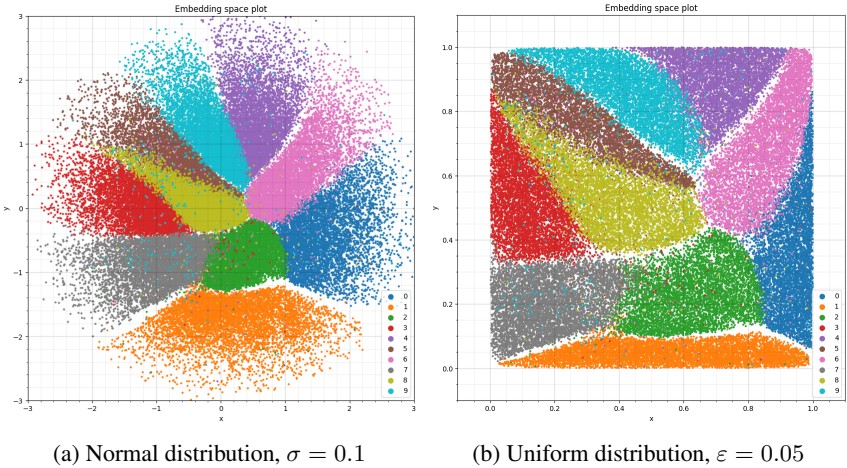

(a) Normal distribution, $\sigma = 0.1$          (b) Uniform distribution, $\varepsilon = 0.05$

Figure 2: Visualization of two-dimensional representations of the MNIST handwritten digits dataset.

Table 1: Linear probing: top-1 & top-5 accuracy (in %) on CIFAR-10/100 under noise injection.

|  | CIFAR-10 | | CIFAR-100 | |
| --- | --- | --- | --- | --- |
|  | top-1 | top-5 | top-1 | top-5 |
| SimCLR | 90.83 | 99.76 | 65.64 | 89.91 |
| SimCLR $\sigma = 0.1$ | 90.96 | 99.72 | 67.03 | 90.49 |
| SimCLR $\sigma = 0.3$ | 91.56 | 99.77 | 65.72 | 89.76 |
| SimCLR $\sigma = 0.5$ | 90.51 | 99.74 | 65.58 | 89.56 |
| VICReg | 90.63 | 99.67 | 65.71 | 88.96 |
| VICReg $\sigma = 0.1$ | 91.09 | 99.68 | 68.92 | 90.50 |
| VICReg $\sigma = 0.3$ | 90.75 | 99.61 | 67.31 | 89.89 |
| VICReg $\sigma = 0.5$ | 91.02 | 99.75 | 66.52 | 89.66 |

Table 2: Linear probing: top-1 & top-5 accuracy (in %) on ImageNet under noise injection (VICReg).

| $\sigma$ | ImageNet-100 | | ImageNet-1k | |
| --- | --- | --- | --- | --- |
|  | top-1 | top-5 | top-1 | top-5 |
| 0 | $72.18 \pm 0.40$ | $92.02 \pm 0.12$ | $67.41 \pm 0.17$ | $87.43 \pm 0.08$ |
| 0.05 | $72.27 \pm 0.38$ | $91.99 \pm 0.18$ | $67.29 \pm 0.20$ | $87.47 \pm 0.06$ |
| 0.1 | $72.07 \pm 0.27$ | $91.65 \pm 0.13$ | $67.30 \pm 0.13$ | $87.43 \pm 0.02$ |
| 0.2 | $71.68 \pm 0.50$ | $91.61 \pm 0.24$ | $67.19 \pm 0.12$ | $87.32 \pm 0.09$ |

An intuition behind the observed slight accuracy drop lies in that noise injection limits the bandwidth of representations, an effect similarly observed in (Lavoie et al., 2022) and resolved by increasing the dimensionality of representations. For a fair comparison with the base method, we keep the original dimensionality (512 and 2048 for ResNet-18 and ResNet-50, respectively).

**Generation** Generative adversarial networks (Goodfellow et al., 2014) produce samples from the underlying latent distribution, which is usually Gaussian. A common approach to make the generation not purely random is to introduce a conditioning vector during the generation (Mirza & Osindero, 2014; Larsen et al., 2016). However, the distribution of the conditioning vector is usually unknown except for specific cases, which hinders the unconditional (or partially conditional) generation via the same model. Thus, a model converting data to the corresponding conditioning vectors with known prior distribution is of great use. In Appendix D, we plug our encoder into a conditional GAN and perform conditional and unconditional generation.

## 6 DISCUSSION

In this paper, we have proposed and thoroughly investigated a novel and efficient approach to the problem of distribution matching of learned representations. Our technique falls into a well-established family of Deep InfoMax self-supervised representation learning methods, and does not require solving min-max optimization problems or employing generative models "post-hoc" to match the distributions. The proposed approach is grounded in the information theory, which allows for a rigorous theoretical justification of the method. In our work, we also explore the possibility of applying our technique to other popular methods for unsupervised and self-supervised representation learning. Consequently, we assert that the proposed approach can be utilized in conjunction with any other method for SSRL, provided that it can be formulated in terms of mutual information or entropy maximization.

To assess the quality of the representations yielded by our method, we (a) visualize the embeddings and run normality tests, and (b) solve a set of downstream tasks in various experimental setups. The results indicate the following:

1. Increasing the noise magnitude facilitates better distribution matching, but only up to a certain point; beyond that, the entire representation learning process begins to deteriorate due to insufficient information being transmitted through the noisy channel.

2. Experiments with MNIST and low-dimensional embeddings indicate the existence of a moderate trade-off between the magnitude of the injected noise and the quality in downstream classification tasks. However, experiments with other infomax-related methods for SSRL and higher embedding dimensionality suggest this influence being negligible.

3. Embeddings acquired via the proposed method can be used to condition generative models, allowing both conditional and unconditional generation due to the distribution of the conditioning vector being known.

**Future work**   As for further research, we consider elaborating on the dual formulation of infomax-based distribution matching. Theorem 4.1 suggests that Gaussian DM can be achieved through a specific choice of the critic network $T(x, y)$. We plan extending this result to other distributions. Additionally, we consider conducting further experiments with generative models to comprehensively assess the quality of conditional and unconditional generation, utilizing conditioning vectors obtained via the proposed method.

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

## A    COMPLETE PROOFS

**Theorem 2.1** (Theorem 8.6.5 in Cover & Thomas (2006))**.** *Let $X$ be a $d$-dimensional absolutely continuous random vector with probability density function $p$, mean $m$ and covariance matrix $\Sigma$. Then*

$$h(X) = h\left(\mathcal{N}(m, \Sigma)\right) - \mathrm{D_{KL}}\left(p \,\|\, \mathcal{N}(m, \Sigma)\right), \qquad h\left(\mathcal{N}(m, \Sigma)\right) = \frac{1}{2}\log\left((2\pi e)^d \det\Sigma\right),$$

*where $\mathcal{N}(m, \Sigma)$ is a Gaussian distribution of mean $m$ and covariance matrix $\Sigma$.*

*Proof of Theorem 2.1.* As for any $m \in \mathbb{R}^d$ it holds $h(X - m) = h(X)$, let us consider a centered random vector $X$. Denoting the probability density function of $\mathcal{N}(0, \Sigma)$ by $\phi_\Sigma$, we have

$$\mathrm{D_{KL}}\left(p \,\|\, \mathcal{N}(0, \Sigma)\right) = \int_{\mathbb{R}^d} p(x) \log \frac{p(x)}{\phi_\Sigma(x)}\, dx = -h(X) - \int_{\mathbb{R}^d} p(x) \log \phi_\Sigma(x)\, dx.$$

Now, consider the second term:

$$\int_{\mathbb{R}^d} p(x) \log \phi_\Sigma(x)\, dx = \mathrm{const} + \frac{1}{2}\, \mathbb{E}_X\, X^T \Sigma^{-1} X =$$

$$= \mathrm{const} + \frac{1}{2}\, \mathrm{Tr}(\Sigma^{-1}\, \mathbb{E}_X[XX^T]) = \mathrm{const} + \frac{1}{2}\, \mathrm{Tr}(\Sigma^{-1}\Sigma) = \mathrm{const} + \frac{d}{2} =$$

$$= \mathrm{const} + \frac{1}{2}\, \mathbb{E}_{\mathcal{N}(0,\Sigma)}\, X^T \Sigma^{-1} X = \int_{\mathbb{R}^d} \phi_\Sigma(x) \log \phi_\Sigma(x)\, dx.$$

Here, in the second line, the cyclic property of the trace is used.  $\square$

**Theorem 2.2.** *Let $X$ be an absolutely continuous random vector with probability density function $p$ and $\mathrm{supp}\, X \subseteq S$, where $S$ has finite and non-zero Lebesgue measure $\mu(S)$. Then*

$$h(X) = h\left(\mathrm{U}(S)\right) - \mathrm{D_{KL}}\left(p \,\|\, \mathrm{U}(S)\right), \qquad h\left(\mathrm{U}(S)\right) = \log \mu(S),$$

*where $\mathrm{U}(S)$ is a uniform distribution on $S$.*

*Proof of Theorem 2.2.*

$$\mathrm{D_{KL}}\left(p \,\|\, \mathrm{U}(S)\right) = \int_S p(x) \log \frac{p(x)}{1/\mu(S)}\, dx =$$

$$= -h(X) + \int_S p(x) \log \mu(S)\, dx = -h(X) + \log \mu(S) = -h(X) + h(\mathrm{U}(S)).$$

$\square$

**Lemma 2.4.** *Let $X$ and $Z$ be random vectors of the same dimensionality, independent under the conditioning vector $Y$. Then*

$$h(X + Z \mid Y) = h(Z \mid Y) + I(X; X + Z \mid Y) \geq h(Z \mid Y),$$

*with equality if and only if there exists a measurable function $g$ such that $X = g(Y)$.*[3]

*Proof of Lemma 2.4.* By the definition of mutual information between $Z$ and $X + Z$ conditioned on $Y$, we have

$$I(X; X + Z \mid Y) = h(X + Z \mid Y) - h(X + Z \mid X, Y),$$

where $h(X + Z \mid X, Y) = h(Z \mid Y)$ due to the independence of $X$ and $Z$ given $Y$.

Now, let us prove that $I(X; X + Z \mid Y) = 0$ if and only if $\exists g : X = g(Y)$. Firstly, if $X = g(Y)$,

$$I(X; X + Z \mid Y) = I(g(Y); g(Y) + Z \mid Y) = I(0; Z \mid Y) = I(0; Z) = 0$$

---

[3]Hereinafter, when comparing random variables, we mean equality or inequality "almost sure".

Next, consider $I(X; X + Z \mid Y) = 0$. Thus, $X$ is conditionally independent both of $X + Z$ and $Z$, which allows for the multiplicative property of characteristic functions to be used:

$$\mathbb{E}\left(e^{i\langle t,Z\rangle} \,\Big|\, Y\right) = \mathbb{E}\left(e^{i\langle t,-X+(Z+X)\rangle} \,\Big|\, Y\right) = \mathbb{E}\left(e^{-i\langle t,X\rangle} \,\Big|\, Y\right) \mathbb{E}\left(e^{i\langle t,Z\rangle} \,\Big|\, Y\right) \mathbb{E}\left(e^{i\langle t,X\rangle} \,\Big|\, Y\right) =$$

$$= \left[\mathbb{E}\left(e^{i\langle t,X\rangle}\right)\right]^* \mathbb{E}\left(e^{i\langle t,X\rangle}\right) \mathbb{E}\left(e^{i\langle t,Z\rangle}\right) = \left|\mathbb{E}\left(e^{i\langle t,X\rangle} \,\Big|\, Y\right)\right|^2 \mathbb{E}\left(e^{i\langle t,Z\rangle} \,\Big|\, Y\right)$$

Note that there exists $\varepsilon(Y) > 0$ such that $\mathbb{E}\left(e^{i\langle t,Z\rangle} \,\big|\, Y\right) \neq 0$ for $\|t\| < \varepsilon(Y)$ (non-vanishing property). Consequently, $\left|\mathbb{E}\left(e^{i\langle t,X\rangle} \,\big|\, Y\right)\right| = 1$ holds for $\|t\| < \varepsilon(Y)$. This implies that the conditional distribution of $X$ given $Y$ is a $\delta$-distribution (Theorem 6.4.7 in (Chung, 2001)), leading to $X = \mathbb{E}(X \mid Y)$, where $g(Y) \triangleq \mathbb{E}(X \mid Y)$ is a $\sigma(Y)$-measurable function. $\quad\square$

**Lemma 3.1.** *Consider the following Markov chain of absolutely continuous random vectors:*

$$f(X) + Z \longrightarrow X \longrightarrow X' \longrightarrow f(X'),$$

*with $Z$ being independent of $(X, X')$. Then*

$$I(f(X'); f(X) + Z) = h(f(X) + Z) - h(Z) - I(f(X) + Z; f(X) \mid f(X')). \quad (5)$$

*Proof of Lemma 3.1.* From the definition of mutual information, we have

$$I(f(X'); f(X) + Z) = h(f(X) + Z) - h(f(X) + Z \mid f(X')).$$

We apply Lemma 2.4 to rewrite the second term

$$h(f(X) + Z \mid f(X')) = h(Z \mid f(X')) + I(f(X); f(X) + Z \mid f(X'))$$
$$= h(Z) + I(f(X); f(X) + Z \mid f(X')),$$

where the independence of $Z$ and $f(X')$ is used. $\quad\square$

**Lemma 3.3.** *Under the conditions of Lemma 3.1, let $\mathbb{P}(X = X' \mid X) \geq \alpha > 0$. Then, $I(f(X) + Z; f(X) \mid f(X')) = 0$ precisely when $f$ is weakly invariant to $X \to X'$.*

*Proof of Lemma 3.3.* According to Lemma 2.4, $I(f(X) + Z; f(X) \mid f(X')) = 0$ if and only if $f(X) = g(f(X'))$ for some $g$. Thus, weak invariance implies $I(f(X) + Z; f(X) \mid f(X')) = 0$.

On the other hand, if $f(X) = g(f(X'))$,

$$\mathbb{P}(f(X) \in (g^{-1} \circ f)(X) \mid X) \geq \mathbb{P}(f(X) = f(X') \mid X) \geq \mathbb{P}(X = X' \mid X) \geq \alpha > 0.$$

Note that the predicate $P(X) \triangleq$ "$f(X) \in (g^{-1} \circ f)(X)$" is not random when conditioned on $X$. Thus, $\mathbb{P}(P(X) \mid X) = \phi(X)$, where $\phi$ is a function taking values in $\{0, 1\}$. As $\phi(X) \geq \alpha > 0$, $\phi(X) = 1$. This implies $g(f(X)) = f(X)$ almost surely.

$\quad\square$

**Theorem 3.4** (Gaussian distribution matching). *Let the conditions of Lemma 3.3 be satisfied. Assume $Z \sim \mathcal{N}(0, \sigma^2 \mathrm{I})$, $\mathbb{E} f(X) = 0$ and $\mathrm{Var}\left(f(X)_i\right) = 1$ for all $i \in \{1, \ldots, d\}$. Then, the mutual information $I(f(X'); f(X) + Z)$ can be upper bounded as follows*

$$I(f(X'); f(X) + Z) \leq \frac{d}{2} \log\left(1 + \frac{1}{\sigma^2}\right), \quad (6)$$

*with the equality holding exactly when $f$ is weakly invariant and $f(X) \sim \mathcal{N}(0, \mathrm{I})$. Moreover,*

$$\mathrm{D_{KL}}\left(f(X) \,\|\, \mathcal{N}(0, \mathrm{I})\right) \leq I(Z; f(X) + Z) - I(f(X'); f(X) + Z) - d \log \sigma.$$

*Proof of Theorem 3.4.* Applying the result from Lemma B.1 gives us:

$$\mathrm{D_{KL}}\left(f(X) + Z \,\|\, \mathcal{N}(0, (1+\sigma^2)\mathrm{I})\right) \leq \frac{d}{2} \log\left(1 + \frac{1}{\sigma^2}\right) - I(f(X'); f(X) + Z).$$

Since KL-divergence is non-negative, we obtain the desired bound. Furthermore, equality holds exactly when $f$ is weakly invariant and $f(X) \sim \mathcal{N}(0, \mathrm{I})$, as discussed in the proof of Lemma B.1.

The second inequality involving $\mathrm{D_{KL}}\left(f(X) \,\|\, \mathcal{N}(0, \mathrm{I})\right)$, follows from Corollary B.3. $\quad\square$

**Theorem 3.5** (Uniform distribution matching). *Under the conditions of Lemma 3.1, let $Z \sim \mathrm{U}([-\varepsilon; \varepsilon]^d)$ and $\mathrm{supp}\, f(X) \subseteq [0; 1]^d$. Then, the mutual information $I(f(X'); f(X) + Z)$ can be upper bounded as follows*

$$I(f(X'); f(X) + Z) \leq d \log \left( 1 + \frac{1}{2\varepsilon} \right), \tag{7}$$

*with the equality if and only if $1/\varepsilon \in \mathbb{N}$, $f$ is weakly invariant, and $f(X) \sim \mathrm{U}(A)$, where the set $A = \{0, 2\varepsilon, 4\varepsilon, \dots, 1\}$ contains $(1/(2\varepsilon) + 1)$ elements.*

*Moreover,*

$$\mathrm{D}_{\mathrm{KL}} \left( f(X) \,\|\, \mathrm{U}([0; 1]^d) \right) \leq I(Z; f(X) + Z) - I(f(X'); f(X) + Z) - d \log(2\varepsilon).$$

*Proof of Theorem 3.5.* Proceeding similarly to the Gaussian case, we apply Lemma B.4, yielding

$$\mathrm{D}_{\mathrm{KL}} \left( f(X) + Z \,\|\, \mathrm{U}([-\varepsilon; 1 + \varepsilon]^d) \right) \leq d \log \left( 1 + \frac{1}{2\varepsilon} \right) - I(f(X'); f(X) + Z),$$

where the left-hand side is non-negative, so we obtain the claimed inequality. The conditions for equality are also established through Lemma B.4.

To prove the second upper-bound, concerning $\mathrm{D}_{\mathrm{KL}} \left( f(X) \,\|\, \mathrm{U}([0; 1]^d) \right)$, it suffices to use the result from Corollary B.6. $\qquad \square$

**Theorem 4.1** (Dual form of Gaussian distribution matching). *Under the conditions of Theorem 3.4,*

$$I(f(X'); f(X) + Z) \geq \mathbb{E}_{\mathbb{P}^+} \left[ T^*_{\mathcal{N}(0, \sigma^2 \mathrm{I})} \right] - \log \mathbb{E}_{\mathbb{P}^-} \left[ \exp \left( T^*_{\mathcal{N}(0, \sigma^2 \mathrm{I})} \right) \right],$$

$$T^*_{\mathcal{N}(0, \sigma^2 \mathrm{I})}(x, y) = \frac{\|y\|^2}{2(1 + \sigma^2)} - \frac{\|y - x\|^2}{2\sigma^2} = \frac{1}{\sigma^2} \left( \langle x, y \rangle - \frac{\|x\|^2 + \|y\|^2/(1 + \sigma^2)}{2} \right),$$

*with the equality holding precisely when $f$ is weakly invariant and $f(X) \sim \mathcal{N}(0, \mathrm{I})$.*

*Proof of Theorem 4.1.* By Remark 2.5 we know that the equality holds if and only if $T^*_{\mathcal{N}(0, \sigma^2 \mathrm{I})}(x, y) = \mathrm{PMI}_{f(X'), f(X) + Z}(x, y) + \mathrm{const}$.

Now, for independent $Y \sim \mathcal{N}(0, \mathrm{I})$ and $Z \sim \mathcal{N}(0, \sigma^2 \mathrm{I})$

$$\mathrm{PMI}_{Y, Y+Z}(x, y) = \log p_{Y+Z|Y}(y \mid x) - \log p_{Y+Z}(y) =$$

$$= \frac{\|y\|^2}{2(1 + \sigma^2)} - \frac{\|y - x\|^2}{2\sigma^2} + \frac{d}{2} \log \left( 1 + \frac{1}{\sigma^2} \right) =$$

$$= T^*_{\mathcal{N}(0, \sigma^2 \mathrm{I})}(x, y) + I(Y; Y + Z).$$

Thus, the equality holds if and only if $I(f(X'); f(X) + Z) = \frac{d}{2} \log(1 + 1/\sigma^2)$, which, in turn, holds precisely when $f$ is weakly invariant and $f(X) \sim \mathcal{N}(0, \mathrm{I})$ (see Theorem 3.4). $\qquad \square$

# B  SUPPLEMENTARY RESULTS ON DISTRIBUTION MATCHING

In this section, we explore the connection between the proposed infomax objective and the problem of distribution matching, formulated in terms of the Kullback-Leibler divergence.

## GAUSSIAN DISTRIBUTION MATCHING

**Lemma B.1.** *Assume the conditions of Theorem 3.4 are satisfied, then for Gaussian distribution matching, we have*

$$\mathrm{D}_{\mathrm{KL}} \left( f(X) + Z \,\|\, \mathcal{N}(0, (1 + \sigma^2)\mathrm{I}) \right) \leq \frac{d}{2} \log \left( 1 + \frac{1}{\sigma^2} \right) - I(f(X'); f(X) + Z),$$

*with equality holding exactly when $f$ is weakly invariant and $f(X) \sim \mathcal{N}(0, \mathrm{I})$.*

*Proof.* From Lemma 3.1 we obtain

$$I(f(X'); f(X) + Z) = h(f(X) + Z) - h(\mathcal{N}(0, \sigma^2 \mathrm{I})) - I(f(X) + Z; f(X) \mid f(X')).$$

Using Theorem 2.1, we can rewrite the first term, which yields

$$\begin{aligned} I(f(X'); f(X) + Z) = {} & h(\mathcal{N}(m, \Sigma)) - h(\mathcal{N}(0, \sigma^2 \mathrm{I})) \\ & - \mathrm{D_{KL}}\left(f(X) + Z \,\|\, \mathcal{N}(m, \Sigma)\right) - I(f(X) + Z; f(X) \mid f(X')), \end{aligned}$$

where $m$ and $\Sigma$ are the mean and covariance matrix of $f(X) + Z$.

To bound the KL-divergence, note that the conditional mutual information is non-negative:

$$\mathrm{D_{KL}}\left(f(X) + Z \,\|\, \mathcal{N}(m, \Sigma)\right) \leq h(\mathcal{N}(m, \Sigma)) - h(\mathcal{N}(0, \sigma^2 \mathrm{I})) - I(f(X'); f(X) + Z).$$

Equality holds exactly when $I(f(X) + Z; f(X) \mid f(X')) = 0$, which is equivalent to $f$ being weakly invariant (see Lemma 3.3).

Next, we estimate the difference between the entropies by observing that

$$h(\mathcal{N}(m, \Sigma)) \leq \sum_{i=1}^{d} h(\mathcal{N}(m_i, \mathrm{Var}(f(X)_i) + \sigma^2)) = d \cdot h(\mathcal{N}(0, 1 + \sigma^2)),$$

with the equality holding if and only if $\Sigma$ is diagonal, which implies $\Sigma = \mathrm{I}$ since $\mathrm{Var}(f(X)_i) = 1$ for all $i \in \{1, \ldots, d\}$. Finally,

$$d \cdot h(\mathcal{N}(0, 1 + \sigma^2)) - h(\mathcal{N}(0, \sigma^2 \mathrm{I})) = \frac{d}{2}\left[\log(1 + \sigma^2) - \log \sigma^2\right] = \frac{d}{2}\log\left(1 + \frac{1}{\sigma^2}\right),$$

which proves the claimed inequality. $\qquad\square$

**Lemma B.2.** *Under the conditions of Theorem 3.4, the following holds:*

$$\begin{aligned} \mathrm{D_{KL}}\left(f(X) + Z \,\|\, \mathcal{N}(0, (1 + \sigma^2)\mathrm{I})\right) \leq {} & \mathrm{D_{KL}}\left(f(X) \,\|\, \mathcal{N}(0, \mathrm{I})\right) = \\ = {} & \mathrm{D_{KL}}\left(f(X) + Z \,\|\, \mathcal{N}(0, (1 + \sigma^2)\mathrm{I})\right) + I(Z; f(X) + Z) - \frac{d}{2}\log\left(1 + \sigma^2\right). \end{aligned}$$

*Proof.* The left-hand inequality follows directly from the data processing inequality for the Kullback-Leibler divergence (Theorem 2.17 in (Polyanskiy & Wu, 2024)).

To establish the right-hand side, we first apply Theorem 2.1, and then use the independence of $f(X)$ and $Z$, yielding $h(f(X)) = h(f(X) + Z \mid Z)$. Thus, we have

$$\mathrm{D_{KL}}\left(f(X) \,\|\, \mathcal{N}(0, \mathrm{I})\right) = h(\mathcal{N}(0, \mathrm{I})) - h(f(X) + Z \mid Z).$$

Next, by using the definition of mutual information and again applying Theorem 2.1, one can write

$$\begin{aligned} h(f(X) + Z \mid Z) = {} & h(f(X) + Z) - I(f(X) + Z; Z) = \\ = {} & h(\mathcal{N}(0, (1 + \sigma^2)\mathrm{I})) - \mathrm{D_{KL}}\left(f(X) + Z \,\|\, \mathcal{N}(0, (1 + \sigma^2)\mathrm{I})\right) - I(f(X) + Z; Z). \end{aligned}$$

Finally, substituting this into the expansion for $\mathrm{D_{KL}}\left(f(X) \,\|\, \mathcal{N}(0, \mathrm{I})\right)$, and noting that $h(\mathcal{N}(0, \mathrm{I})) - \mathcal{N}(0, (1 + \sigma^2)\mathrm{I})) = -\frac{d}{2}\log(1 + \sigma^2)$ completes the proof. $\qquad\square$

**Corollary B.3.** *In the Gaussian distribution matching setup (Theorem 3.4), we have*

$$\mathrm{D_{KL}}\left(f(X) \,\|\, \mathcal{N}(0, \mathrm{I})\right) \leq I(Z; f(X) + Z) - I(f(X'); f(X) + Z) - d \log \sigma.$$

*Proof.* This follows directly from combining Lemma B.1 and Lemma B.2. $\qquad\square$

UNIFORM DISTRIBUTION MATCHING

**Lemma B.4.** *Let the conditions of Theorem 3.5 hold, then the following bound applies for uniform distribution matching:*

$$D_{KL}\left(f(X) + Z \parallel U([-\varepsilon; 1 + \varepsilon]^d)\right) \leq d\log\left(1 + \frac{1}{2\varepsilon}\right) - I(f(X'); f(X) + Z),$$

*with the equality if and only if $1/\varepsilon \in \mathbb{N}$, $f$ is weakly invariant, and $f(X) \sim U(A)$, where the set $A = \{0, 2\varepsilon, 4\varepsilon, \dots, 1\}$ contains $(1/(2\varepsilon) + 1)$ elements.*

*Proof.* Similarly to the proof of Lemma B.1, we use the decomposition of the infomax objective from Lemma 3.1. Afterward, we apply Theorem 2.2 to the term $h(f(X) + Z)$:

$$
\begin{aligned}
I(f(X'); f(X) + Z) = {} & h(U([-\varepsilon; \varepsilon + 1]^d)) - h(U([-\varepsilon; \varepsilon]^d)) \\
& - D_{KL}\left(f(X) + Z \parallel U([-\varepsilon; \varepsilon + 1]^d)\right) - I(f(X) + Z; f(X) \mid f(X')).
\end{aligned}
$$

Therefore, the KL-divergence can be bounded as:

$$
\begin{aligned}
D_{KL}\left(f(X) + Z \parallel U([-\varepsilon; \varepsilon + 1]^d)\right) & \leq h(U([-\varepsilon; \varepsilon + 1]^d)) - h(U([-\varepsilon; \varepsilon]^d)) - I(f(X'); f(X) + Z) \\
& = d\log\left(1 + \frac{1}{2\varepsilon}\right) - I(f(X'); f(X) + Z),
\end{aligned}
$$

with equality achieved if and only if the mutual information between $f(X) + Z$ and $f(X)$ conditioned on $f(X')$ is zero, which occurs precisely when $f$ is weakly invariant (see Lemma 3.3).

Next, we show when the equality holds. The probability density function of the sum of independent random vectors $f(X)$ and $Z$ is given by the convolution:

$$\int_{\mathbb{R}^d} p_{f(X)}(x) p_Z(z - x) dx = \prod_{i=1}^{d} \int_{z_i - \varepsilon}^{z_i + \varepsilon} \left(\frac{1}{1/(2\varepsilon) + 1} \sum_{a \in A} \delta(z_i - a)\right) \frac{dx_i}{2\varepsilon} = \frac{1}{(1 + 2\varepsilon)^d} \int_{[-\varepsilon; 1 + \varepsilon]^d} dx.$$

Here, we used the independence of the components, with $Z_i$ uniformly distributed on $[-\varepsilon, \varepsilon]$ and $f(X)_i$ uniformly distributed over the discrete set $A = \{0, 2\varepsilon, 4\varepsilon, \dots, 1\}$.

Therefore, $(f(X) + Z) \sim U([-\varepsilon; 1 + \varepsilon]^d)$. Given this, one can calculate the mutual information explicitly:

$$I(f(X); f(X) + Z) = h(f(X) + Z) - h(Z) = d\log\left(1 + \frac{1}{2\varepsilon}\right).$$

This concludes the proof. $\qquad\square$

**Lemma B.5.** *Under the conditions of Theorem 3.5, the following holds:*

$$D_{KL}\left(f(X) \parallel U([0; 1]^d)\right) = D_{KL}\left(f(X) + Z \parallel U([-\varepsilon; \varepsilon + 1]^d)\right) + I(Z; f(X) + Z) - d\log(1 + 2\varepsilon).$$

*Proof.* One can build on the reasoning from Lemma B.2 by applying Theorem 2.2 to express the KL-divergence between $f(X)$ and $U([0; 1]^d)$ as follows:

$$D_{KL}\left(f(X) \parallel U([0; 1]^d)\right) = h(U([0; 1]^d)) - h(f(X) + Z \mid Z) = -h(f(X) + Z \mid Z).$$

Using Theorem 2.2 again, the conditional entropy can be expressed as

$$h(f(X) + Z \mid Z) = h(U([-\varepsilon; \varepsilon + 1]^d)) - D_{KL}\left(f(X) + Z \parallel U([-\varepsilon; \varepsilon + 1]^d)\right) - I(f(X) + Z; Z).$$

To conclude it remains to calculate $h(U([-\varepsilon; \varepsilon + 1]^d)) = d\log(1 + 2\varepsilon)$. $\qquad\square$

**Corollary B.6.** *In the uniform distribution matching setup (Theorem 3.5), we have*

$$D_{KL}\left(f(X) \parallel U([0; 1]^d)\right) \leq I(Z; f(X) + Z) - I(f(X'); f(X) + Z) - d\log(2\varepsilon).$$

*Proof.* Applying Lemma B.4 alongside Lemma B.5 is enough. $\qquad\square$

## C  DETAILS OF IMPLEMENTATION

For experiments on MNIST dataset, we use a simple ConvNet with three convolutional and two fully connected layers. A three-layer fully-connected perceptron serves as a critic network for the InfoNCE loss. We provide the details in Table 3. We use additive Gaussian noise with $\sigma = 0.6$ as an input augmentation. Training hyperparameters are as follows: batch size = 1024, 2000 epochs, Adam optimizer (Kingma & Ba, 2017) with learning rate $10^{-3}$.

Table 3: The NN architectures used to conduct the tests on MNIST images in Section 5.

| NN | Architecture | |
| --- | --- | --- |
| ConvNet, $24 \times 24$ images | ×1: | Conv2d(1, 32, ks=3), MaxPool2d(2), BatchNorm2d, LeakyReLU(0.01) |
| | ×1: | Conv2d(32, 64, ks=3), MaxPool2d(2), BatchNorm2d, LeakyReLU(0.01) |
| | ×1: | Conv2d(64, 128, ks=3), MaxPool2d(2), BatchNorm2d, LeakyReLU(0.01) |
| | ×1: | Dense(128, 128), LeakyReLU(0.01), Dense(128, dim) |
| Critic NN, pairs of vectors | ×1: | Dense(dim + dim, 256), LeakyReLU(0.01) |
| | ×1: | Dense(256, 256), LeakyReLU(0.01), Dense(256, 1) |

The results on CIFAR datasets (Krizhevsky, 2009) in Table 1 were obtained with the standard configuration of SSL methods. Namely, we use ResNet-18 (He et al., 2015) backbone. Projection head for SimCLR consists of two linear layers, for VICReg – 3 layers. Respective configurations are [2048, 256] and [2048, 2048, 2048], meaning embedding dimensions are 256 and 2048, respectively. We apply a standard set of augmentations:

```
PretrainTransform: Compose(
    RandomResizedCrop(
        size=(32, 32),
        scale=(0.08, 1.0),
        ratio=(0.75, 1.3333333333333333),
        interpolation=InterpolationMode.BICUBIC,
        antialias=True
    )

    RandomApply(
        ColorJitter(
            brightness=(0.6, 1.4),
            contrast=(0.6, 1.4),
            saturation=(0.8, 1.2),
            hue=(-0.1, 0.1)
        )
    )

    RandomGrayscale(p=0.2)
    GaussianBlur(p=0.0)
    Solarization(p=0.0)
    RandomHorizontalFlip(p=0.5)
    ToTensor()
    Normalize(
    mean=[0.4914, 0.4822, 0.4465],
    std=[0.247, 0.2435, 0.2616],
    inplace=False)
)
TestTransform: Compose(
        ToTensor()
        Normalize(
          mean=[0.4914, 0.4822, 0.4465],
          std=[0.247, 0.2435, 0.2616],
          inplace=False
```

)
)

Training hyperparameters are as follows: batch size 256, 800 epochs, LARS optimizer (You et al., 2017) with clipping, base learning rate 0.3, momentum 0.9, trust coefficient 0.02, weight decay $10^{-4}$. For SimCLR, we use temperature 0.2, for VICReg – standard hyperparameters $(25, 25, 1)$.

The source code is available on GitHub repository.

## D  CONDITIONING GENERATIVE ADVERSARIAL NETWORKS

In this section, we leverage our method to generate conditioning vectors for a conventional cGAN setup (Mirza & Osindero, 2014). We use the two-dimensional Gaussian embeddings of the MNIST dataset, acquired from the noise level $\sigma = 0.1$. For conditioned generation, we get embeddings from a batch of original images. For unconditioned generation, embeddings are sampled from $\mathcal{N}(0, \mathrm{I})$. The results are presented in Figures 3 and 4.

## E  ADDITIONAL EXPERIMENTS ON CIFAR-10

Similarly to embedding MNIST into two-dimensional space, we run experiments with CIFAR-10. We use a ResNet-18 architecture augmented with an additional linear layer to map the 512-dimensional output to $\mathbb{R}^2$, along with a batch normalization layer without learnable parameters (i.e., 'affine=False'). We experimented with both contrastive (SimCLR) and non-contrastive (VICReg) objectives, observing similar results across both methods. The experimental pipeline closely follows the theoretical setup:

- Unaugmented Input: The unaugmented image $X$ is fed to the encoder, yielding $f(X)$, followed by noise injection to obtain $f(X) + Z$.
- Augmented Input: The augmented image $X'$ is fed to the encoder to obtain $f(X')$, without noise injection.

Both outputs are then processed by the projection head and loss function (e.g., InfoNCE). The noise $Z$ is sampled from a normal distribution with standard deviation $\sigma$. We conducted pre-training across various noise magnitudes with $\sigma \in \{0.0, 0.01, 0.025, 0.05, 0.1, 0.2, 0.2658, 0.5\}$. For each $\sigma$, we ran training with 5 different random seeds with `base_lr = 0.1` for 400 epochs (visualized in Figure 5). To speed up convergence we additionally trained 3 models ($\sigma \in \{0, 0.05, 0.1\}$) with `base_lr = 0.5` for 800 epochs (visualized in Figure 6).

After pre-training, we evaluated downstream performance using ROC AUC and performed normality tests on the learned 2D representations from a total of 40 trained models (that were trained for 400 epochs) (see Figure 7), depicting similar results to our previous MNIST experiments (Figure 1).

## F  ADDITIONAL EXPERIMENTS ON IMAGENET

To better assess the accuracy-DM trade-off in complex setups, additional experiments with the ImageNet datasets are conducted with VICReg loss.

For ImageNet-100, as backbone we use ResNet-18 followed by batch normalization layer without learnable parameters, i.e. `BatchNorm1d(512, affine=False)`. Projection head consists of standard MLP sequence: `nn.Linear(512, 2048)`, `nn.BatchNorm1d(2048)`, `nn.ReLU()`, `nn.Linear(2048, 2048)`, `nn.BatchNorm1d(2048)`, `nn.ReLU()`, `nn.Linear(2048, 2048)`.

For ImageNet-1k, we use ResNet-50 as backbone architecture, followed by batch normalization layer without learnable parameters (`dim=2048`). The projection head is comprised similar to the above setup of 3-layer MLP with hidden dimension = 8192.

In both setups, we run pre-training for 100 epochs and conduct linear probing afterwards closely following standardized protocol for augmentation and training. for ImageNet-1k, batch size and

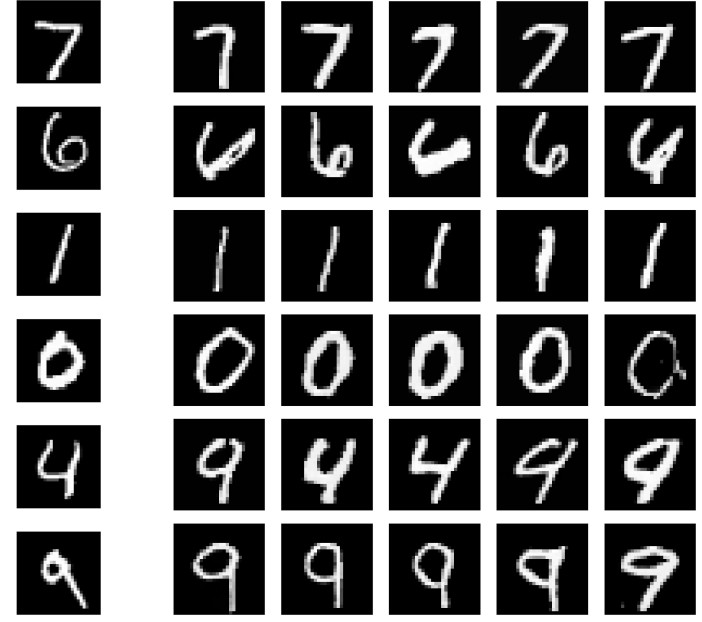

(a) Encoded images        (b) Generated images

Figure 3: Results of conditional generation.

Figure 4: Results of unconditional generation.

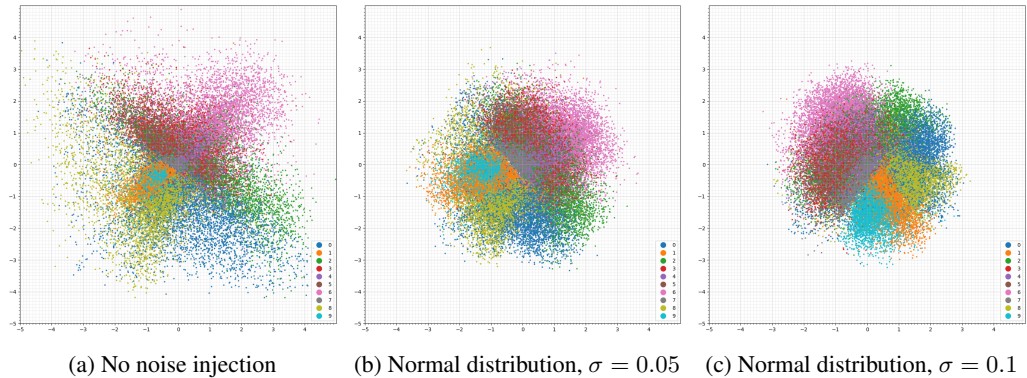

(a) No noise injection     (b) Normal distribution, $\sigma = 0.05$     (c) Normal distribution, $\sigma = 0.1$

Figure 5: Visualization of 2D representations of the CIFAR-10 dataset, lr=0.1, 400 epochs.

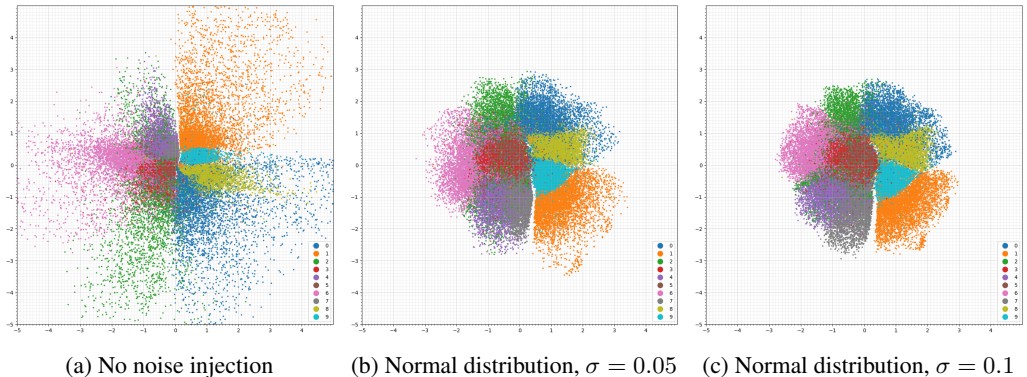

(a) No noise injection     (b) Normal distribution, $\sigma = 0.05$     (c) Normal distribution, $\sigma = 0.1$

Figure 6: Visualization of 2D representations of the CIFAR-10 dataset, lr=0.5, 800 epochs.

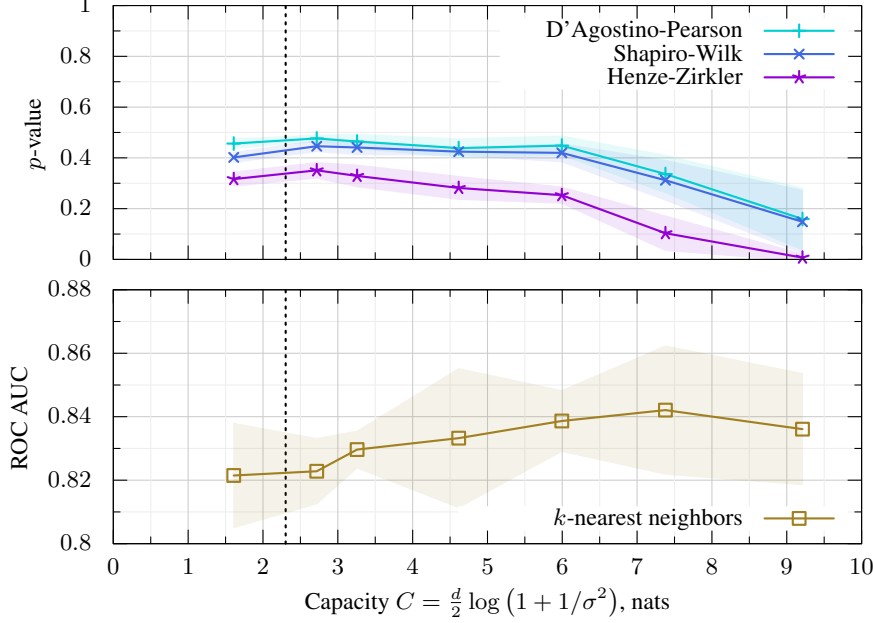

Figure 7: Results for CIFAR10 dataset in the Gaussian DM setup for $d = 2$ with varying capacity $C = \frac{d}{2} \log \left(1 + 1/\sigma^2\right)$, measured in nats (units of information based on natural logarithms). The dotted line denotes the minimal capacity required to preserve the information about the class labels in $f(X) + Z$. We run 5 experiments for each point and report mean values and 99% asymptotic confidence intervals. InfoNCE loss is used to approximate (2).

other (e.g. optimizer, data augmentation) hyperparameters set to specified in the original paper Bardes et al. (2021), however, evaluation batch size is set to 2048 for faster results. For ImageNet-100, batch size = 256, optimizer's trust coefficient = 0.02, base learning rate = 0.5, weight decay = `1e-4`, inspired by da Costa et al. (2022). The resulting accuracies are reported in Table 2 across 5 random seeds.

