# OpenReview forum: "Efficient Distribution Matching of Representations via Noise-Injected Deep InfoMax"
_ICLR.cc/2025/Conference — ICLR 2025 Poster_

### Official Review · Reviewer_RR5m · 2024-10-25

**Soundness:** 3
**Presentation:** 4
**Contribution:** 3
**Rating:** 6
**Confidence:** 3

**Summary:**

The paper modifies the Deep InfoMax approach to learning representations that match specific distributions (uniform and normal). Specifically, it suggests adding noise to the normalized representations encoded from the inputs, before training them to maximize the mutual information with encoded input augmentations. It provides theoretical upper bounds for the mutual information between the noisy representation and the representation of the augmentation. It shows empirical results demonstrating the distribution matching, and the trade-off regarding the amounts of noise - a high noise level facilitates better distribution matching, but might result in worse representation learning.

**Strengths:**

1. The paper is very clear and well-written. It is easy to follow the derivations and the main ideas.

2. It provides a good theoretical and empirical analysis of the method and the trade-offs between representation learning and distribution matching.

3. The presented modification is simple, and yet it provides a non-negligible improvement in usefulness (sampling).

**Weaknesses:**

1. The quality of conditional and unconditional sampling, the main property that can be achieved from a distribution matching is not tested on larger scale data. The paper can significantly improve with generation results on a larger scale of data (e.g. ImageNet/Cifar100).

2. Even for the representation learning tests, this paper can improve with testing on larger datasets. The results of Figure 1, for example, are mostly saturated on MNIST and it will be useful to verify the same on larger more diverse datasets.

3. While the paper has a section that argues that the same method can be applied to other SSRLs, it lacks empirical results that can demonstrate the usefulness/fragility of the training regime with noise.

**Questions:**

My main questions are about scaling:

Will the representation learning improve over the baseline InfoMax in more conventional large-scale settings - ImageNet linear-probing/fine-tuning?

How stable is the training with noise under large-scale settings?

For generation and sampling - can you show interpretable individual directions in the learned representation distribution (again - on more advanced datasets like FFHQ/ImageNet)?

---

> ### Author Response · Authors · 2024-11-21
>
> Dear Reviewer **RR5m,**
>
> Thank you for your valuable feedback and insightful comments. We have carefully addressed your concerns and provided updates based on new experiments and analysis.
>
> ### **Conditional and Unconditional Sampling**
> We are currently finalizing ImageNet-1K pre-training and conducting experiments on conditional and unconditional sampling for CIFAR-10. While we are eager to evaluate the proposed approach in practical generative settings, we emphasize that this experiment is auxiliary to the main contribution of the paper. In our submission, generation is considered as a supplementary validation effort, but conducting a comprehensive analysis requires considerable computational resources. We will provide an update with results as soon as these experiments are complete.
>
> ###  **Representation Learning**
> We extended our representation learning experiments to CIFAR-10, analogous to the MNIST experiment in the original submission.
>
> **Experimental setup**. We use a ResNet-18 architecture augmented with a linear layer (mapping 512-dimensional output to $\mathbb{R}^2$) and a batch normalization layer without learnable parameters (`affine=False`). Both contrastive (SimCLR) and non-contrastive (VICReg) objectives are tested, showing consistent results.
> The experimental pipeline closely follows the theoretical setup:
> - Unaugmented Input: The unaugmented image $X$ is fed to the encoder, yielding $f(X)$, followed by noise injection to obtain $f(X) + Z$.
> - Augmented Input: The augmented image $X'$ is fed to the encoder to obtain $f(X')$, without noise injection.
>
> The outputs are then processed by the projection head and loss function (e.g., InfoNCE). Noise $Z$ is sampled from a normal distribution with standard deviation $\sigma$.
>
> Models were trained for 400 epochs across $\sigma \in {0.0, 0.01, 0.025, 0.05, 0.1, 0.2, 0.2658, 0.5}$ with five random seeds and base learning rate `base_lr=0.1`. To accelerate convergence, we also trained models with $\sigma \in {0, 0.05, 0.1}$ for 800 epochs with `base_lr=0.5`. Downstream performance was evaluated using ROC AUC, with normality tests on the learned 2D representations. Results are shown in the updated manuscript's Appendix E and F (Figures 5–7).
> Figures 5 and 6 demonstrate that CIFAR-10 embeddings into 2D space increasingly resemble a Gaussian distribution as $\sigma$ increases, consistent with MNIST results. The overall results demonstrate the approach generalizes to more complex datasets than MNIST.
>
> #### **Q1: Performance in Large-Scale Settings**
>
> We acknowledge the original submission lacked experiments on large-scale datasets due to computational constraints. Motivated by your feedback and that of others, we conducted experiments on ImageNet-100 (a 100-class subset of ImageNet-1K) and are awaiting completion of full ImageNet-1K training.
>
> We used VICReg with $\sigma \in {0.0, 0.05, 0.1, 0.2}$ and a ResNet-18 encoder (output dimension = 512). Models were trained for 100 epochs with standard hyperparameters across 5 random seeds. Performance is reported below (also in Appendix F, Table 3):
> | $\sigma$ | Top-1 Accuracy (%) | Top-5 Accuracy (%) |
> | :-------- | :------------------: | :------------------: |
> | 0.0      | 72.18 ± 0.40       | 92.02 ± 0.12       |
> | 0.05     | 72.27 ± 0.38       | 91.99 ± 0.18       |
> | 0.1      | 72.07 ± 0.27       | 91.65 ± 0.13       |
> | 0.2      | 71.68 ± 0.50       | 91.61 ± 0.24       |
>
> Noise injection minimally affects performance for small $\sigma$, suggesting the method scales well to larger datasets. Full ImageNet-1K results will be shared once training is complete.
> #### **Q2: Stability in Large-Scale Settings**
> Training on ImageNet-100 revealed sensitivity to higher noise magnitudes ($\sigma = 0.2$), with top-1 accuracy showing slight degradation and increased variance. However, training remains robust with small noise injection, and no collapse was observed across all runs.
>
> ### **Transferability to Other SSRL Methods**
> This is a fair concern as there are a great number of SSRL methods, e.g. contrastive, non-contrastive, self-distillation, clusterisation-based approaches to name a few.  In our work, we focused on representative methods from contrastive (SimCLR) and non-contrastive (VICReg) families. SimCLR’s InfoNCE loss closely aligns with the InfoMax principle underlying our theoretical setup. VICReg also has information-theoretic interpretations, making it a suitable candidate for empirical validation. As for the broader applicability, given the connections between joint-embedding methods in existing literature, we expect our approach to generalize to similar SSRL methods. Our current experiments serve as representative case studies.
>
> Thank you once again for your constructive feedback! We look forward to your further comments and suggestions.

---

> > ### Comment · Reviewer_RR5m · 2024-11-22
> >
> > I would like to thank the authors for answering my questions and concerns. I encourage the authors to include the results for ImageNet and ImageNet100 in the next revision. I agree that the sampling quality is tangential to the paper's main contribution.
> >
> > After reading the rebuttal and looking at the additional results in this response and the response for y7PL, I decided to keep my score.

---

> > > ### Author Response · Authors · 2024-11-25
> > >
> > > Dear Reviewer **RR5m**,
> > >
> > > Thank you again for your work! The results for ImageNet100 and ImageNet, as well as all the required information to reproduce the experiments, are included in the current revision. Please, refer to Appendix F.
> > >
> > > The resulting accuracies are reported in Table 3 across 5 random seeds for ImageNet-100 and on 1 seed on ImageNet-1k due to compute limitations. We hope the other runs to finish before the discussion period ends. However, even now one could cautiously argue that the effects of the noise injection on the downstream accuracy are negligible across all four datasets we tested.

---

### Official Review · Reviewer_UXaV · 2024-10-31

**Soundness:** 3
**Presentation:** 3
**Contribution:** 2
**Rating:** 6
**Confidence:** 3

**Summary:**

This paper proposes an improved Deep Infomax (DIM) method for self-supervised representation learning. Overall, an independent noise $Z$ is added to the normalized representations of the data. The authors demonstrate that the method is effective since it keeps the same training objective as the DIM. Experimental results also show the effectiveness.

**Strengths:**

1. The research has been well accomplished since both the theoretical effectiveness and experimental results are provided.

2. The background and fundamental method are well described, so readers can easily follow the idea.

**Weaknesses:**

The only improvement in this work is the additive noise in DIM, which may be not very overwhelming. However, if this small improvement can be thoroughly discussed and analyzed, the contribution of this work could still be significant. Therefore, my concerns mainly focus on the discussions and analysis:

1. Since the authors claim in the title that the proposed method is an efficient matching method, it should be clarified why the improved approach is efficient. However, I only see that adding noise will not impact the DIM objective while improving the performance. The efficiency is not discussed.

2. The advantage of adding noise is also not well discussed. It is stated that Lemma 3.1 highlights the importance of both additive noise and input data augmentation. Still, I only see the analysis of the significance of augmentation and do not understand why adding noise is also necessary.

**Questions:**

If the authors can clarify the significance of the additive noise in the theoretical view, this work will be complete, in my opinion. Then, I will consider changing my rating.

---

> ### Author Response · Authors · 2024-11-17
>
> Dear Reviewer **UXaV**, thank you sincerely for reviewing our article! In the following text, we provide responses to your concerns. We hope that all of them are addressed properly.
>
> **Weaknesses:**
>
> 1. Acquiring embeddings conforming to a specific distribution (i.e., distribution matching, DM) is an auxiliary, yet important task in representation learning due to several applications which we list on lines 50-55 (generative modelling, statistical analysis, disentanglement, outliers detection). Conventional approaches to DM are either imprecise (lines 56-60), or almost as expensive as full-fledged generative modelling, requiring a setup similar to GANs, normalizing flows or diffusion models (lines 60-68). In contrast, we propose achieving DM via normalization and noise injection only, which is cheap (as no additional NN is required), and justified both theoretically (Section 3) and experimentally (Section 5). Thus, we claim our method to be efficient.
> 2. We highlight the importance of noise injection in Theorems 3.4-3.5 and Remark 3.6. Note that without the noise injection, the bounds from the aforementioned theorems deteriorate, becoming $+\infty$, which renders the maximum entropy properties inapplicable. Thus, one can not be assured that maximization of DIM objective yields normally or uniformly distributed embeddings, even if the normalization is still performed. Moreover, note that without the noise injection, the $I(f(X) + Z; f(X) \mid f(X'))$ term in Lemma 3.1 is either $0$ (when $f$ is weakly invariant to $ X \to X' $), or $+\infty$ (otherwise), rendering this decomposition useless for the later analysis.
>
>    Additional intuition on the importance of noise injection can be acquired from line 818 in the proof of Lemma 3.1:
>
>    $$ I(f(X');f(X) + Z) = h(f(X) + Z) - h(f(X) + Z \mid f(X')) $$
>
>    Note that if $Z = 0$, and some kind of normalization is still employed (so $h(f(X))$ is upper-bounded), $h(f(X) \mid f(X'))$ can get arbitrary small, up to $-\infty$ for weakly invariant $f$. This makes the minimization of this term preferable to the maximization of $h(f(X))$ during the gradient descent (it is even possible that $I(f(X');f(X)) \to +\infty$, but $h(f(X))$ stays far from the maximal value). However, if $Z$ is non-zero, $h(f(X) + Z \mid f(X'))$ saturates, with the limiting value being lower for higher noise-to-signal ratio. Thus, the magnitude of the noise serves as a trade-off between the distribution matching and the weak invariance. This explanation is also backed by the experimental results from Figure 1, which illustrate the aforementioned trade-off.
>
> **Questions:** in the response above, we stress out the importance of the noise injection from the theoretical point of view. In short, the desired distribution matching through DIM and embedding normalization only is not achievable; injecting the noise is crucial.
>
> This clarification will be added to the upcoming revision of our manuscript. We hope that you will find our answer satisfactory. If there are some parts of our response which require further clarification, please, let us know.
>
> We again sincerely thank Reviewer **UXaV** for carefully reading our article and pointing out the parts of our theoretical framework which require additional explanation.

---

> > ### Comment · Reviewer_UXaV · 2024-11-23
> >
> > Thanks to the authors for the responses. My concerns have been addressed. For the first problem, I suggest the authors clarify the given analysis and conclusion in the paper at a proper location.

---

> > > ### Author Response · Authors · 2024-11-25
> > >
> > > Dear Reviewer **UXaV**,
> > >
> > > Thank you again for the feedback! The clarifications regarding the first problem have been added to the paragraph on lines 69-73, where we contrast our method with other approaches.

---

### Official Review · Reviewer_y7PL · 2024-11-02

**Soundness:** 2
**Presentation:** 3
**Contribution:** 3
**Rating:** 6
**Confidence:** 4

**Summary:**

The paper propose to add gaussian or uniform noise to the output of the encoder in contrastive methods.

**Strengths:**

The paper defines the notation and robustly use it throughout the paper making it easy to follow along as a    reader. I appreciate the clarity and reading the method and the experimental section. I was also pleasantly  surprised of the novelty of the idea proposed, while being very simple.

**Weaknesses:**

The main weakness of the paper is that the experiments are conducted on simple and sometimes trivial           benchmarks. The raw input space of MNIST can already be well represented using t-SNE [1] and thus any SSL      representation learning experiments performed on this dataset are dubious. I encourage the authors to          reproduce the visualization obtained on MNIST with CIFAR-10 or CIFAR-100.

One potential issue with adding noise to the output representation is that we limit the bandwith of the        representation. Therefore, while we may see improvement on small datasets such as CIFAR-100, this improvement  may vanish as we scale up the size of the dataset. I encourage the authors to consider running their           experiments on ImageNET to validate the their method scales up to larger and more diverse datasets.

[Minor] There are some issues with the formatting. E.g. in page 1, the citation overflows to the second page   breaking the reference.

[1] https://www.kaggle.com/code/venkatkrishnan/t-sne-decomposition-on-mnist-dataset

**Questions:**

* Is the linear probe trained on the output of the encoder or on some intermediate layer?
* Are the gains observed in Table 1 transferrable when training on ImageNet?

---

> ### Author Response · Authors · 2024-11-21
> **Additional experiments on CIFAR-10 (for 2d embedding) and ImageNet-100**
>
> Dear Reviewer y7PL,
>
> Thank you for your thoughtful review and valuable feedback. We are pleased to address your concerns below.
>
> ## **Weaknesses**:
>
> ### **2D Experiment on CIFAR-10**
>
> We appreciate your observation regarding the triviality of embedding MNIST into 2D space. Our primary goal with the MNIST experiment was to validate that the representations align with the theoretical predictions for noise injection. To address your suggestion and validate our approach on a more complex dataset, we have conducted additional experiments on CIFAR-10.
>
> In these experiments, we used a ResNet-18 architecture augmented with an additional linear layer to map the 512-dimensional output to $\mathbb{R}^2$, along with a batch normalization layer without learnable parameters (i.e., `affine=False`). We experimented with both contrastive (SimCLR) and non-contrastive (VICReg) objectives, observing similar results across both methods. The experimental pipeline closely follows the theoretical setup:
>
> 1. **Unaugmented Input:** The unaugmented image $X$ is fed to the encoder, yielding $f(X)$, followed by noise injection to obtain $f(X) + Z$.
> 2. **Augmented Input:** The augmented image $X'$ is fed to the encoder to obtain $f(X')$, without noise injection.
>
> Both outputs are then processed by the projection head and loss function (e.g., InfoNCE). The noise $Z$ is sampled from a normal distribution with standard deviation $\sigma$. We conducted pre-training across various noise magnitudes with $\sigma \in \{0.0, 0.01, 0.025, 0.05, 0.1, 0.2, 0.2658, 0.5\}$, and for each $\sigma$, we ran training with 5 different random seeds with `base_lr=0.1` for 400 epochs (visualized in the updated Appendix E, Figure 5)  To speed up convergence we additionally trained 3 models ($\sigma \in \{0, 0.05, 0.1\}$) with `base_lr=0.5` for 800 epochs (visualized in Appendix E, Figure 6).
>
> After pre-training, we evaluated downstream performance using ROC AUC and performed normality tests on the learned 2D representations (see Appendix F, Figure 7), analogous to our previous MNIST experiments (Figure 1). Consistent with the MNIST results, the CIFAR-10 samples were successfully embedded into 2D space, exhibiting a distribution that increasingly resembles a Gaussian as $\sigma$ increases (Figures 5 and 6). This demonstrates that our approach generalizes to more complex datasets than MNIST.
>
> ### **Large-Scale Datasets**
> We acknowledge that the original submission lacked experiments on large-scale datasets due to computational constraints. Motivated by your feedback and that of other reviewers, we have secured additional resources to conduct experiments on larger datasets. While we await the completion of full ImageNet-1K training, we present preliminary results on ImageNet-100, a 100-class subset of ImageNet-1K.
>
> Using the VICReg objective with noise injection, we experimented with $\sigma \in \{0.0, 0.05, 0.1, 0.2\}$ (note that $\sigma=0$ corresponds to the original VICReg setup), employing a ResNet-18 encoder (output dim = 512) and standard hyperparameters. We trained the models for 100 epochs and evaluated the downstream performance of the learned representations (encoder outputs). The results are summarized in the table below (Appendix F, Table 3):
> | $\sigma$      | Top-1 Accuracy (%) | Top-5 Accuracy (%) |
> |:---------------|:------------------:|:------------------:|
> | 0.0           | 72.18 ± 0.40       | 92.02 ± 0.12       |
> | 0.05          | 72.27 ± 0.38       | 91.99 ± 0.18       |
> | 0.1           | 72.07 ± 0.27       | 91.65 ± 0.13       |
> | 0.2           | 71.68 ± 0.50       | 91.61 ± 0.24       |
>
> These results indicate that noise injection has a negligible impact on performance for small noise magnitudes, suggesting that our method scales to larger and more diverse datasets without significant degradation in accuracy. Once training finishes on full ImageNet-1K, we will update the table.
>
> ## **Questions**:
>
> 1. **Is the linear probe trained on the output of the encoder or on some intermediate layer?**
>
>    The linear probe is trained on the output of the encoder. This approach is standard for classification benchmarks in self-supervised learning methods within the vision domain, particularly when using ResNet-type encoders. Most methods, such as SimCLR, VICReg, Barlow Twins, BYOL, and SimSiam, follow this pattern.
>
> 2. **Are the gains observed in Table 1 transferable when training on ImageNet?**
>
>    While we are awaiting the completion of full ImageNet-1K pre-training, we can provide preliminary insights into the transferability of our method based on the ImageNet-100 experiments mentioned above. The results suggest that our approach maintains comparable performance on larger datasets. Please refer to the table above (also included in the updated manuscript) for detailed results.
>
> We hope that these additional experiments and clarifications address your concerns. Thank you again for your constructive feedback, which has helped us strengthen our work.

---

> > ### Comment · Reviewer_y7PL · 2024-11-21
> > **Thank you**
> >
> > I appreciate that the authors have taken the time to run the additional experiments. I am satisfied with the CIFAR experiments and I will raise my score to a 6.
> >
> > Regarding the ImageNet experiments, I believe that an issue might be that size of the output vector is too small since the bandwidth of the representation is constrained by the additive noise. Some prior works [1] found that over-complete representations led to stronger performance when they induced sparsity (which is another way to limit the bandwidth) on the representation. Perhaps, you will find the same in your work.
> >
> > [1] https://arxiv.org/abs/2204.00616

---

> > > ### Author Response · Authors · 2024-11-27
> > >
> > > Reviewer **y7PL**,
> > >
> > > Thank you very much for your insights regarding the slight drop in accuracy we observe on the ImageNet dataset when using higher noise magnitudes! We share your intuition regarding this effect. Indeed, additive noise, as well as the SEM normalization explored in [1], may limit the bandwidth of the representations. One can expect this effect to be negated by increasing the dimensionality.
> > >
> > > As we use a standardized protocol for evaluating SSRL methods (see, e.g., [2,3]), current dimensionality is also adopted from the literature to make our results comparable with other works. However, motivated by your remark, we will provide additional discussion of the results on ImageNet through the lenses of the aforementioned hypothesis in the next revision.
> > >
> > > [1] https://arxiv.org/abs/2204.00616
> > >
> > > [2] Susmelj, I., Heller, M., Wirth, P., Prescott, J., Ebner, M., & et al. Lightly [Computer software]. https://github.com/lightly-ai/lightly
> > >
> > > [3] Victor Guilherme Turrisi da Costa, Enrico Fini, Moin Nabi, Nicu Sebe, and Elisa Ricci. sololearn: A library of self-supervised methods for visual representation learning

---

### Official Review · Reviewer_Vzb6 · 2024-11-03

**Soundness:** 3
**Presentation:** 3
**Contribution:** 3
**Rating:** 8
**Confidence:** 3

**Summary:**

This work introduces  noise injection to shape a self-supervised learning represenation as Gaussian or Uniformed representation. In short, by injecting a Guassian noise and restricting the convariance of f(x), f(x) tends to be gaussian distirbutioned. By injecting an uniformed distribution within a narrow range $[-\epsilon, \epsilon]$ and restricting the range of f(x) (e.g. within [0,1]), f(x) tends to be uniformly distributioned.

The reason for the gaussian distribution comes from the fact that the Gussian distribution has the max entropy of all distributions with a fixed covariance. The reason for the uniform distribution comes from uniform distribution U(a, b) has the max entropy of a distribution within range [a,b].

Apart from the theorical analysis, this work also present interesting plots to demenstrate the two kinds of learned representation, i.e. Gaussian and Uniformed.

**Strengths:**

- The intuition of two noise injection, Gaussian and Uniform, methods are resonable. The reason for the gaussian distribution comes from the fact that the Gussian distribution has the max entropy of all distributions with a fixed covariance. The reason for the uniform distribution comes from uniform distribution U(a, b) has the max entropy of a distribution within range [a,b].

- This work provides a theorical proof of the effect of these two noise injections.

- This paper provides rich background knowledges.

- The experiments in this paper (cifar, mnist) demenstrate the usefulness of the two noise injection methods.

**Weaknesses:**

- line 236-239 explains why injecting a gaussian noise helps learn a gaussian representation. This sentence is helpful for understanding. It would be good to have a similar short explaination before the proof of uniform noise injection. For example, the reason of small $\epsilon$, discrete support $A$, and the entropy of uniformed distribution.

- Large experiments beyond cifar and mnist are helpful to convince readers.

**Questions:**

- what is the meaning of "nats" in Figure 1 caption "Capacity, nats"? Does it refer to $C = d/2 log(1 + 1 / \sigma^2)$?

---

> ### Author Response · Authors · 2024-11-20
> **Official Comment by Authors**
>
> Dear Reviewer Vzb6,
>
> Thank you for your comprehensive review and for taking the time to carefully evaluate our submission. Below, we provide detailed responses to your comments and questions, and we hope that all your concerns are properly addressed.
>
> **Weaknesses**:
> 1. We will add a concise clarification to this section, similar to the one provided for Gaussian noise, to enhance clarity. Below we provide the text to be included in the upcoming revision of our article.
>
>    We consider $ Z $ distributed uniformly on $ [-\varepsilon; \varepsilon]^d $, and assume that $ f(X) $ has bounded support on $ [0; 1]^d $. By Theorem 2.2, the entropy $ h(f(X) + Z) $ reaches its maximum precisely when $ f(X) + Z $ is uniformly distributed. While the independence of $ Z $ and $ f(X) $ does not imply that $ f(X) $ is uniform, the distribution of $ f(X) $ that maximizes $ h(f(X) + Z) $ is a discrete uniform distribution over a finite grid within $ [0; 1]^d $. As $ \varepsilon $ approaches zero, the distribution of $ f(X) $ converges to a continuous uniform distribution over $ [0; 1]^d $. Thus, injecting uniform noise asymptotically drives $ f(X) $ towards a uniform representation over $ [0; 1]^d $. This reasoning is formalized in Theorem 3.5
>
> 2. Thank you for your suggestion. We are already in the process of running our model on larger datasets, specifically ImageNet, and hope to include these results before the end of the rebuttal period.
>
> **Questions**:
> Thank you for pointing out the term "nats" in Figure 1. The quantity $ C = \frac{d}{2} \log \left( 1 + 1/\sigma^2 \right) $, derived in Theorem 3.4, represents the capacity mentioned in the caption of Figure 1. It is measured in 'nats', units of information based on natural logarithms. We will update the Figure 1 caption to explicitly define "nats" and also include the mathematical expression for the capacity in the axis label.
>
> We sincerely thank you once again for your constructive feedback, which has been valuable in refining our work. If any further concerns arise, we will be glad to address them.

---

> ### Author Response · Authors · 2024-11-25
> **Awaiting your reply**
>
> Dear Reviewer **Vzb6**,
>
> Once again, thank you very much for your detailed review. As we are nearing the end of the discussion period, we would like to ask if the concerns you raised have been addressed.
>
> To remedy the first weakness, a concise clarification was added; please, refer to our response or to the beginning of Section 3.2 in our revised manuscript.
>
> To address the second weakness, we've updated the manuscript to include experiments on larger scale datasets (including 2d embedding and representation learning with noise injection on CIFAR and ImageNet datasets; please see Appendix E and F in the updated manuscript). For your convenience, the resulting accuracies are also reported below; the experiments were conducted across 5 random seeds for ImageNet-100 and on 1 seed on ImageNet-1k due to compute limitations. We hope the other runs to finish before the discussion period ends.
>
> | $\sigma$  | Top-1 Accuracy (%) (ImageNet-100) | Top-5 Accuracy (%) (ImageNet-100) | Top-1 Accuracy (%) (ImageNet) | Top-5 Accuracy (%) (ImageNet) |
> | :-------- | :------------------: | :------------------: | :------------------: | :------------------: |
> | 0.0       | 72.18 ± 0.40         | 92.02 ± 0.12         | 67.57                | 87.54                |
> | 0.05      | 72.27 ± 0.38         | 91.99 ± 0.18         | 67.33                | 87.54                |
> | 0.1       | 72.07 ± 0.27         | 91.65 ± 0.13         | 67.21                | 87.42                |
> | 0.2       | 71.68 ± 0.50         | 91.61 ± 0.24         | 67.17                | 87.30                |
>
> Overall, the additional experiments are in line with the results obtained in the original empirical study.
>
> We hope this update addresses your concerns, we thank you again for your review and look forward to your further comments and suggestions. We understand that the discussion period is short, and we sincerely appreciate your time and help!

---

> ### Comment · Reviewer_Vzb6 · 2024-11-26
>
> Thank authors for the explaination and additional experiments on ImageNet. My concerns are addressed and I would like to raise my score.
>
> Here is a summary of my suggestions:
>
> 1) give an (intuitive) explaination about how uniform representation is learned (gaussian representation is well explained).
>
> 2) move ImageNet experiment into main text, because it is more convincing.

---

> ### Author Response · Authors · 2024-11-27
> **Official Comment by Authors**
>
> Dear Reviewer **Vzb6**,
>
> Thank you once again for your review and valuable suggestions. We appreciate your positive feedback. Below, we provide responses to your specific suggestions:
>
> 1. We have revised our explanation of how uniform representation is learned as follows:
>
>    Now consider $Z$ distributed uniformly on $[-\varepsilon; \varepsilon]^d$. We assume that $f(X)$ has bounded support within $[0, 1]^d$. This restriction serves as an alternative to fixing the second-order moments, as in the Gaussian case, and is easy to implement in practice (e.g., via a sigmoid function)
>
>    By Theorem 2.2, the entropy $h(f(X) + Z)$ is maximized when $f(X) + Z$ follows a uniform distribution. This can be achieved when $f(X)$ conforms to a discrete uniform distribution over an equidistant grid within $[0, 1]^d$. As $\varepsilon$ approaches zero, the distribution of $f(X)$ gradually converges to a continuous uniform distribution over $[0, 1]^d$. Thus, injecting uniform noise asymptotically drives $f(X)$ toward a uniform representation over $[0, 1]^d$. This reasoning is formalized in Theorem 3.5.
>
>    We will replace the paragraph in the next revision.
>
> 2. The results from linear probing for ImageNet will be included in Table 2 of the main body, with additional details provided in Appendix F.
>
> We are delighted to hear that your concerns have been addressed. We sincerely appreciate the time and effort you have put into reviewing our paper.

---

### Author Response · Authors · 2024-11-22
**New revision**

Dear Reviewers,

We would like to thank you again for the time spent on reading our article and for the profound and constructive reviews you have provided. To address the raised concerns, we answered each reviewer individually. We also revised our manuscript to introduce the requested changes, which we list below:
1. Added the mathematical expression for the capacity to the axis label in Figure 1.
1. Defined the notion of "nats" in the caption of Figure 1 (line 400).
1. Rewrote the first paragraph of Section 3.2 to improve clarity for uniform distribution matching (lines 264–270).
1. Added the 2d representations visualization (Appendix E Figures 5 and 6) and normality testing results (Appendix E Figure 7) for CIFAR-10.
1. Added the results for ImageNet-100 downstream accuracy using the models trained with noise injection (Appendix F Table 3)
1. Added the detailed description of experimental setups accordingly (Appendix E and F).
1. Added the clarification regarding the importance of noise injection (lines 220–221 and 231–324).

---

### Author Response · Authors · 2024-11-30
**General response and revision**

Dear Reviewers,

We sincerely thank you for your thoughtful reviews and valuable feedback on our work. We are grateful for the time and effort each of you dedicated to reviewing our manuscript.

We deeply appreciate your recognition of the rigorous theoretical foundations and empirical validation of our work (Vzb6, UXaV, RR5m). We are also grateful for your acknowledgment of the clarity of our manuscript and the well-structured background (Vzb6, y7PL, UXaV, RR5m). Finally, we are encouraged by your positive remarks on the simplicity and novelty of our approach, noting its valuable contribution (Vzb6, y7PL, RR5m).

Your observations have significantly strengthened our manuscript. In response to your suggestions, we have introduced the following changes into our final revision:

1. Added experiments on larger-scale datasets:
   - Detailed results for CIFAR are included in Appendix E.
   - Accuracies on ImageNet and ImageNet-100 are presented in the main body (Table 2), with further details in Appendix F.
2. Clarified why our method is effective by contrasting it with related approaches (lines 69–73).
3. Revised our explanation of how uniform representations are learned (lines 266–273).
4. Discussed the observed slight drop in accuracy on ImageNet with higher noise magnitudes (lines 429–432).

Thank you once again for your constructive feedback and guidance. We are glad that all the raised concerns have been addressed.

---

### Meta-Review · Area_Chair_QEoo · 2024-12-21

**Metareview:**

This paper introduces a method for enforcing representations learnt with self-supervised learning (specifically Deep InfoMax) to follow a specific distribution by normalizing and injecting appropriate noise into encoder outputs. The method is simple yet novel, and is theoretically grounded while also being shown to be effective in the experiments. The paper further shows connections to commonly used self-supervised learning techniques and empirically validates its effectiveness on these. The paper is well-written and clear. While reviewers had initial concerns about the lack of experiments on larger scale datasets, these were addressed in the rebuttal (see below). Following the rebuttal, all reviewers were positive about the paper and the AC concurs, recommending acceptance.

**Additional Comments On Reviewer Discussion:**

The key concerns raised by reviewers were the lack of experiments on larger-scale datasets beyond MNIST and CIFAR, as well as clarifications on the proposed method and its advantages over existing approaches. In the rebuttal, the authors included additional results on ImageNet and CIFAR-10/CIFAR-100 and included additional clarifications and discussions in the manuscript regarding the method and its advantages. Reviewers were satisfied with the rebuttal and 3 reviewers raised their scores. Reviewers were thus unanimous in recommending acceptance for the paper, which the AC agrees with.

---

### Decision · Program_Chairs · 2025-01-22

Accept (Poster)